# Consumption in the G20 nations causes particulate air pollution resulting in two million premature deaths annually

Keisuke Nansai[1,2✉], Susumu Tohno[3], Satoru Chatani[4], Keiichiro Kanemoto[5], Shigemi Kagawa[6], Yasushi Kondo[7], Wataru Takayanagi[1] & Manfred Lenzen[2]

Worldwide exposure to ambient $PM_{2.5}$ causes over 4 million premature deaths annually. As most of these deaths are in developing countries, without internationally coordinated efforts this polarized situation will continue. As yet, however, no studies have quantified nation-to-nation consumer responsibility for global mortality due to both primary and secondary $PM_{2.5}$ particles. Here we quantify the global footprint of $PM_{2.5}$-driven premature deaths for the 19 G20 nations in a position to lead such efforts. G20 consumption in 2010 was responsible for 1.983 [95% Confidence Interval: 1.685–2.285] million premature deaths, at an average age of 67, including 78.6 [71.5–84.8] thousand infant deaths, implying that the G20 lifetime consumption of about 28 [24–33] people claims one life. Our results indicate that G20 nations should take responsibility for their footprint rather than focusing solely on transboundary air pollution, as this would expand opportunities for reducing $PM_{2.5}$-driven premature mortality. Given the infant mortality footprint identified, it would moreover contribute to ensuring infant lives are not unfairly left behind in countries like South Africa, which have a weak relationship with G20 nations.

[1] Material Cycles Division, National Institute for Environmental Studies, 16-2 Onogawa, Tsukuba 305-8506, Japan. [2] ISA, School of Physics, Faculty of Science, The University of Sydney, Camperdown, NSW 2006, Australia. [3] Graduate School of Energy Science, Kyoto University, Sakyo-ku, Kyoto 606-8501, Japan. [4] Regional Environment Conservation Division, National Institute for Environmental Studies, 16-2 Onogawa, Tsukuba 305-8506, Japan. [5] Research Institute for Humanity and Nature, 457-4 Motoyama, Kamigamo, Kita-ku, Kyoto 603-8047, Japan. [6] Faculty of Economics, Kyushu University, 6-19-1 Hakozaki, Higashi-ku, Fukuoka 812-8581, Japan. [7] Faculty of Political Science and Economics, Waseda University, 1-6-1 Nishi-Waseda, Shinjuku-ku, Tokyo 169-8050, Japan. ✉email: nansai.keisuke@nies.go.jp

To achieve planetary health[1], a world that is healthy for both the global biosphere and human civilization, the critical challenge is to mitigate the human health hazards, as well as suppress the environmental impacts created by socioeconomic activities, reining in the latter to within the Earth's environmental tolerance. Among the many environmental problems affecting human health, the greatest threat is that posed by the inhalation of particles with an aerodynamic diameter of 2.5 μm or less, abbreviated to $PM_{2.5}$[2]. According to the World Health Organization (WHO)[3], in 2016 respiratory and cardiovascular diseases and cancer caused by exposure to ambient $PM_{2.5}$ were responsible for approximately 4.2 million premature deaths, i.e., deaths occurring before the average age of death in the population concerned. The majority of these deaths were in countries with low and middle incomes[3], while 91% of the global population lives in areas where air quality is below WHO guidelines[4]. The loss to the global workforce due to premature deaths attributable to $PM_{2.5}$ was equivalent to 225 billion US dollars in 2013[5]. While the reduction in air pollution due to the COVID-19 pandemic[6] has mitigated human health impacts somewhat, it is neither sufficient nor lasting[7–9].

WHO air quality guidelines[4] stipulate that the annual mean concentration of $PM_{2.5}$ in the atmosphere should not exceed 10 μg/m³. Compared with a concentration of 35 μg/m³, as is frequently observed in developing cities, 10 μg/m³ or less can be expected to decrease $PM_{2.5}$-related deaths by 15%[4]. In China, with the largest number of premature deaths[2], compliance with the WHO guidelines would reduce annual $PM_{2.5}$-related deaths by 81.5%[10]. Atmospheric $PM_{2.5}$ levels can be reduced by cutting emissions of primary $PM_{2.5}$ particles and the precursors of secondary particles, such as sulphur dioxides, nitrogen oxides and volatile organic compounds (VOCs). Key reduction measures include improvements in combustion technology for power generation, transportation and waste treatment[3], as well as flue gas treatment technology for dust collection, denitration, and desulphurization. While our understanding of the impact of $PM_{2.5}$ on human health, target concentrations and abatement technologies has undeniably grown, though, the reality is that developing countries cannot be expected to implement such countermeasures without financial and technical support from the international community. With respect to $PM_{2.5}$, therefore, two critical challenges remain: to understand how responsibility for the high number of premature deaths is distributed across the nations of the world; and to set up a process for deploying countermeasures, embedded in a high-level international policy framework.

By applying consumption-based accounting[11], the health impact of $PM_{2.5}$ induced, by way of the supply chain, by the consumption of one country in another[12–17], that of one region in another[18–22] and that of one racial-ethnic group in another[23] can be calculated; this impact is referred to as a footprint. Multinational footprint analysis[24] helps clarify the rationale and degree of responsibility underlying implementation of joint measures among affecting and affected countries. To date, the multinational footprints of $PM_{2.5}$ primary particle emissions[25] have been quantified and the multinational footprints of their associated health impacts[14–16,26] have been clarified, taking into account the effects of both trade and transboundary pollution. However, only two studies[12,17] on multinational footprints have addressed the health impacts of secondary $PM_{2.5}$ particles formed by chemical reactions in the atmosphere, which in major global cities contribute more to atmospheric $PM_{2.5}$ concentrations than primary black carbon particles[27]. Furthermore, as those two studies have a lower geographical resolution or scope—calculating footprints for each of 13 aggregated regions of the world[12] and for five affluent countries with respect to 34 Asian countries[17]—they fail to address nation-to-nation consumer responsibility for this challenge at the global level.

In order to equitably accommodate consumer responsibility[28,29], joint measures will need to be implemented both bilaterally and multilaterally, to facilitate negotiations among related countries, promote decision-making at a high policy level and initiate international agreements. Such a high-level international agenda will increase the likelihood of such initiatives succeeding. The meeting of the Group of Twenty, or G20, is a regular international meeting attended by leaders of developed countries with a high degree of consumer responsibility and developing countries incurring consumption-related health damage. G20 members represent around 80% of the world's economic output, two-thirds of the global population and three-quarters of international trade[30]. There is a process in which the members of the G20 discuss challenges and measures related to financial and socioeconomic issues. These discussions are then published as outcome documents by the leaders in attendance. The G20 is thus, potentially, a productive high-level meeting that could be used to advance joint measures for resolving the problems associated with $PM_{2.5}$-related health issues.

To date, however, no studies have quantified the consumer responsibility of G20 nations for the substantial health impacts caused by atmospheric $PM_{2.5}$. This lack of scientific knowledge risks delaying international collaborative efforts to safeguard the victims of the $PM_{2.5}$ pollution. To this end, we aim to identify the consumer responsibility of G20 nations for the global premature deaths caused by both primary and secondary $PM_{2.5}$ particles and compare this figure with the premature deaths caused by the domestic and transboundary pollution due to each nation's production-based emissions, the primary focus at present.

## Results

**Footprint of $PM_{2.5}$ mortality induced by G20 nations**. In 2010, 1.983 [95% Confidence Interval (CI): 1.685, 2.285] million premature deaths in five disease categories (Lower Respiratory Infection (LRI), Chronic Obstructive Pulmonary Disease (COPD), Lung Cancer (LC), stroke and Ischemic Heart Disease (IHD)), occurred in 199 countries and regions listed in Supplementary Table 1 in the supporting information (SI) around the world as a result of airborne $PM_{2.5}$ induced by consumption in the 19 nations eligible for the G20 summit presidency (Table 1). The total number of premature deaths due to $PM_{2.5}$ worldwide is estimated to be 4.019 [CI: 3.413, 4.630] million, a figure in broad agreement with previous estimates, although the year, method and data employed differ; e.g., 3.440 million in 2007 for four diseases (COPD, LC, stroke, IHD)[12] and 3.15 million[31] and 3.23 million[32] for the same five diseases in 2010. The greatest number of deaths occur in China: 1.195 [CI: 1.029, 1.362] million, followed by India: 0.907 [CI: 0.790, 1.022] million, while the other studies estimated them at 1.024 million[12] deaths in 2007, 1.367 million[33] and 1.300 million (over 30 years only)[34] deaths in 2013 for the four diseases in China, and 0.9 million[35] deaths for the five diseases in 2014 in India.

Since half the 4.019 million deaths are attributable to consumption in the G20 nations, the G20 meeting is a promising high-level policy platform for formulating an international response to mitigate the health impacts of $PM_{2.5}$ on a global scale. In terms of nations with the largest premature mortality footprint due to $PM_{2.5}$ ($PM_{2.5}$ premature death footprint), China has the largest at 905 [CI: 777, 1033] thousand deaths, followed by India at 493 [CI: 431, 555] thousand, the US at 139 [CI: 112, 168] thousand, Russia at 74.6 [CI: 60.4, 89.4] thousand, and Indonesia at 52.7 [CI: 42.1, 63.4] thousand. Given the differences in year and diseases studied, these death footprints can be deemed

**Table 1 Summary of PM$_{2.5}$ premature death footprint and PM$_{2.5}$ premature deaths due to the production-based emissions of each G20 nation in 2010.**

| G20 nations | PM$_{2.5}$ premature death footprint [1000 deaths/y] | 95% CI (confidence interval) [1000 deaths/y] | Secondary particles contribution to footprint [%] | Share of foreign deaths in footprint [%] | Premature deaths by production-based PM$_{2.5}$ emissions [1000 deaths/y] | 95% CI [1000 deaths/y] | Share of foreign deaths in deaths by production-based emissions [%] | Difference in deaths between footprint and production-based emissions [1000 deaths/y] | 95% CI [1000 deaths/y] |
|---|---|---|---|---|---|---|---|---|---|
| Argentina | 5.66 | [4.62, 6.71] | 52 | 41 | 7.18 | [5.80, 8.58] | 6.4 | −1.5 | [−1.9, −1.2] |
| Australia | 5.74 | [4.61, 6.94] | 67 | 82 | 1.56 | [0.97, 2.22] | 1.1 | 4.2 | [3.6, 4.7] |
| Brazil | 29.8 | [23.6, 36.3] | 43 | 24 | 27.2 | [21.1, 33.4] | 2.7 | 2.6 | [2.4, 2.8] |
| Canada[a] | 13.5 | [10.8, 16.4] | 69 | 85 | 7.96 | [5.79, 10.4] | 56 | 5.5 | [5.0, 6.0] |
| China | 905 | [777, 1033] | 54 | 7.6 | 1090 | [936, 1242] | 5.8 | −185 | [−209, −158] |
| France[a] | 27.9 | [22.8, 33.3] | 73 | 83 | 19.7 | [15.6, 24.1] | 65 | 8.2 | [7.1, 9.2] |
| Germany[a] | 44.0 | [36.0, 52.3] | 76 | 78 | 38.9 | [31.1, 47.3] | 54 | 5.1 | [5.0, 5.1] |
| India | 493 | [431, 555] | 48 | 13 | 549 | [480, 617] | 12 | −56 | [−62, −49] |
| Indonesia | 52.7 | [42.1, 63.4] | 41 | 13 | 59.9 | [47.5, 72.4] | 3.8 | −7.1 | [−8.9, −5.4] |
| Italy[a] | 22.4 | [18.4, 26.5] | 77 | 84 | 12.6 | [10.1, 15.3] | 58 | 9.7 | [8.3, 11] |
| Japan[a] | 41.8 | [34.8, 49.0] | 67 | 74 | 16.5 | [13.2, 19.9] | 9.7 | 25 | [22, 29] |
| Mexico | 14.9 | [12.3, 17.6] | 64 | 39 | 11.8 | [9.6, 14.0] | 9.5 | 3.2 | [2.7, 3.7] |
| Russia | 74.6 | [60.4, 89.4] | 77 | 38 | 76.3 | [61.4, 91.7] | 25 | −1.7 | [−2.3, −1.0] |
| Saudi Arabia | 12.4 | [10.7, 14.] | 84 | 96 | 8.72 | [7.6, 9.8] | 91 | 3.6 | [3.1, 4.3] |
| South Africa | 20.1 | [17.1, 22.9] | 66 | 39 | 23.3 | [19.9, 26.6] | 30 | −3.2 | [−3.6, −2.8] |
| South Korea | 20.2 | [17.0, 23.4] | 71 | 83 | 13.5 | [11.3, 15.8] | 60 | 6.6 | [5.7, 7.6] |
| Turkey | 25.8 | [21.6, 30.2] | 71 | 71 | 22.4 | [18.7, 26.2] | 61 | 3.4 | [2.9, 4.0] |
| United Kingdom[a] | 34.3 | [28.1, 40.7] | 75 | 74 | 20.2 | [15.9, 24.6] | 38 | 14 | [12, 16] |
| United States[a] | 139 | [112, 168] | 72 | 62 | 64.8 | [48, 83] | 9.7 | 75 | [64, 85] |
| G7 total | 323 | [263, 386] | 72 | 62 | 181 | [140, 224] | 22 | 143 | [123, 162] |
| G20 total | 1983 | [1685, 2285] | 57 | 11 | 2071 | [1759, 2384] | 8.4 | −88 | [−99, −74] |

[a]G7 nations

broadly similar to the figures reported earlier by Zhang et al.[12]: China at 835 thousand, India at 442 thousand, the US at 165 thousand and Russia at 74.0 thousand in 2007. With 75 [CI: 64, 85] thousand deaths, the US stands out in terms of the mortality difference between the footprint and production-based emissions, while China is at the opposite end of the spectrum, with 1.090 [CI: 936, 1.242] million premature deaths due to production emissions, which is 185 [CI: 158, 209] thousand higher than for the footprint. Overall, 13 G20 nations have a $PM_{2.5}$ premature death footprint exceeding the premature deaths by their own production emissions (Table 1).

With the exception of Indonesia and India, about half the deaths in the above nations' footprints are attributable to secondary particles, confirming that factoring in the impact of secondary particles is essential to understanding the health impact footprint of $PM_{2.5}$ in the G20 nations. The proportion of foreign deaths in the respective footprint is 7.6% for China, 13% for India, 62% for the US, 38% for Russia, and 13% for Indonesia. Considering the G20 as a whole, there are eleven nations with a percentage of foreign deaths exceeding 50%, while for six of these nations, including Saudi Arabia (96%) and Canada (85%), the figure is over 80%. This highlights the fact that around half the G20 countries need to address their $PM_{2.5}$ footprint as an international issue. What the G20 nations have in common is that the human health footprint of the $PM_{2.5}$ associated with their consumption exceeds the impact on other countries of transboundary $PM_{2.5}$ pollution from their production activities.

While appreciation of the number of premature deaths among G20 nations due to transboundary pollution in the form of production emissions creates a motive to implement joint $PM_{2.5}$ reduction measures among China, Japan, and South Korea, and between the United States and Canada, such motivation is unlikely to arise among countries that are geographically distant, because the mutual direct impact is very small. Quantifying the deaths occurring among nations based on the footprint calculation, on the other hand, enables us to discover closer interactions among them, especially for China and India, which may broadly motivate joint measures to secure $PM_{2.5}$ reduction (Fig. 1).

**Countries with the largest share in the footprint**. To further clarify the bilateral relationships embodied in the footprint, it was broken down by the countries impacted (Fig. 2). For example, the US consumption has a significant impact on China (38.7 [CI: 33.3, 44.1] thousand deaths), India (12.9 [CI: 11.3, 14.5] thousand deaths), Mexico (3.9 [CI: 3.2, 4.6] thousand deaths), and Russia (2.1 [CI: 1.7, 2.5] thousand deaths), as well as on the US itself (52.9

[CI: 39.2, 67.9] thousand deaths). In addition, the US consumer responsibility also extends to non-G20 countries, such as Bangladesh (2.1 [CI: 1.8, 2.3] thousand deaths), and the Philippines (1.5 [CI: 1.2, 1.7] thousand deaths), which are among the top 10 countries affected. This is over and against only 58.5 [CI: 43.4, 75.0] thousand domestic US deaths (close to a previous estimate of 54.9 thousand deaths[31] in 2010), which at present tends to be the sole concern as long as the focus is exclusively on production-based emissions (Supplementary Fig. 1 in the SI). Compared with the situation in 2007[12], the $PM_{2.5}$ impact of US consumption has shifted slightly to India (9.9 thousand in 2007) from China (54.4 thousand in 2007). The footprint of other G20 nations likewise includes impacts on non-G20 nations. Given that it takes time and considerable financial resources for a single nation to develop abatement measures jointly with individual non-G20 nations, it would be beneficial if the G20 nations could work together with nations outside the G20 to reduce premature deaths due to $PM_{2.5}$.

China features prominently in the footprints of Australia, Canada, Japan and South Korea, with China ranking highest among the impacted countries (Fig. 2). In addition to impacting their own countries, France, Germany, Italy, and the United Kingdom (UK) likewise have footprints biased toward China, which perhaps unsurprisingly, is a hotspot for health impacts in the consumption supply chain of developed countries. In the case of production emissions, however, only Korea has a significant impact on China, and China's presence as an affected country is weak for many countries (Supplementary Fig. 1 in the SI). In addition, the G20 nations are less likely to be among the top 10 affected countries in terms of production emissions, as is the case for Australia, the US and Canada. Therefore, in the G20 high-level policy meetings, taking a footprint perspective will make discussions on joint $PM_{2.5}$ measures more constructive.

**Infant deaths and average age of death in the footprint**. The elderly are more susceptible to death than younger age groups, with the highest number of deaths occurring among those aged 80 and over (Fig. 3). However, there is significant mortality among infants aged 0–5; consumption in G20 nations comes at the expense of 78.6 [CI: 71.5, 84.8] thousand infant lives annually, more than in any other age group under 50. Although the average age of death in the footprint due to the total consumption of the G20 nations is 67, certain countries such as South Africa (57 years), Saudi Arabia (59 years), India (60 years) and Indonesia (62 years) show distinctly younger mortality ages owing to the inclusion of high infant death rates. In concrete figures, the

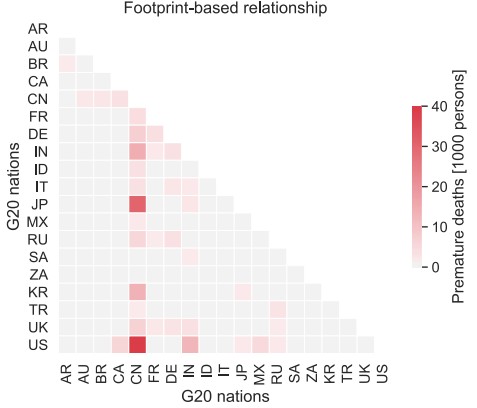
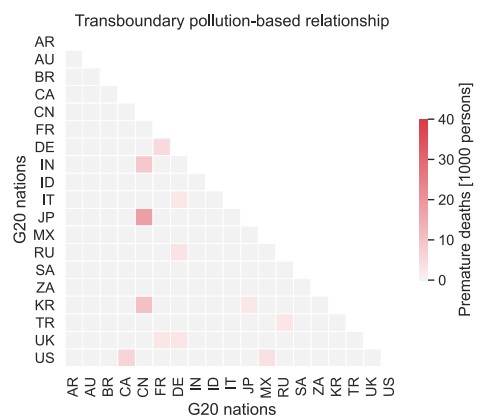

**Fig. 1 Comparison of footprint-based and transboundary pollution-based relationships among G20 nations for the number of $PM_{2.5}$-related premature deaths.** Country codes as follows: Argentina (AR), Australia (AU), Brazil (BR), Canada (CA), China (CN), France (FR), Germany (DE), India (IN), Indonesia (ID), Italy (IT), Japan (JP), Mexico (MX), Russia (RU), Saudi Arabia (SA), South Africa (ZA), South Korea (KR), Turkey (TR), United Kingdom (UK), and United States (US).

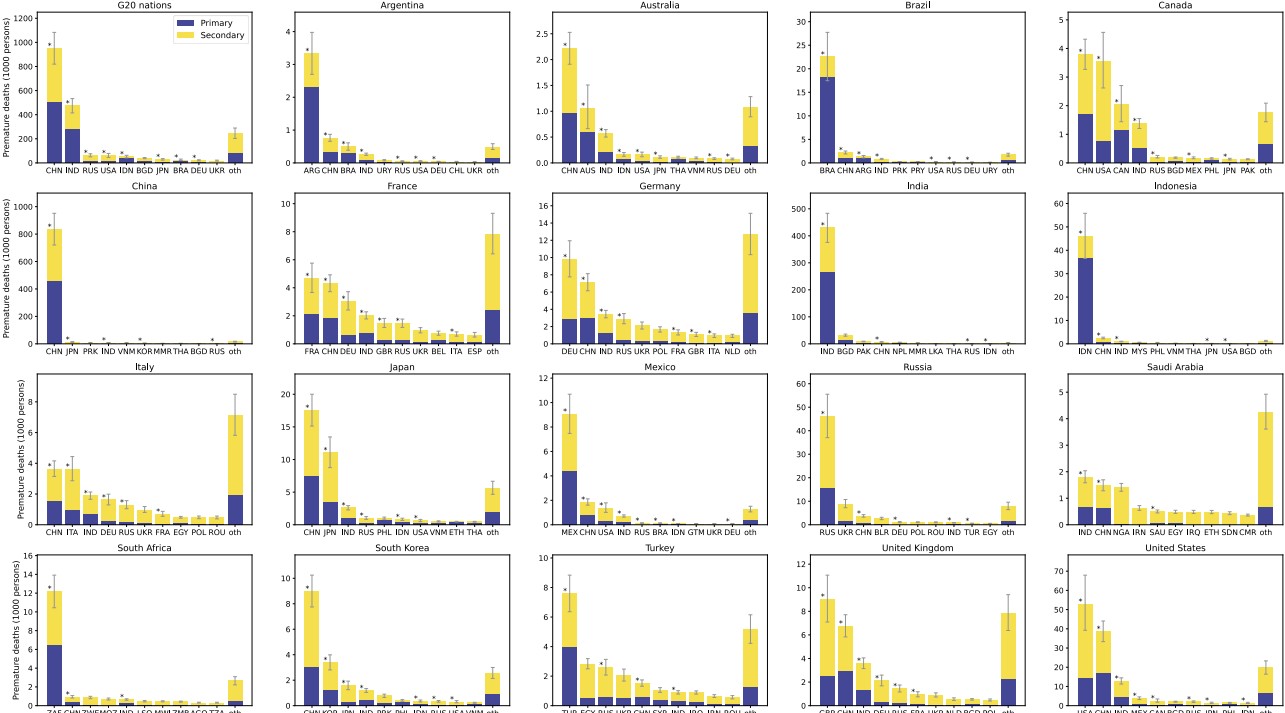

**Fig. 2 Breakdown of PM$_{2.5}$ premature death footprint of G20 nations by impacted country and contribution of primary and secondary particles to footprint.** Asterisks above bars indicate G20 nations with presidency rights. Country and region codes within the top 10 are as listed in Supplementary Table 1 in the Supporting Information; others are aggregated as 'oth'. Error bars indicate 95% confidence intervals. The centre of the bars indicates the mean value.

footprints of consumption in South Africa, Saudi Arabia, India and Indonesia include 2.6 [CI: 2.4, 2.8], 1.7 [CI: 1.6, 1.9], 50.7 [CI: 46.2, 54.7], and 2.1 [CI: 1.9, 2.3] thousand infant deaths, respectively, resulting in a high percentage of such deaths. In addition, although the bias towards infant deaths is not as prominent as in the above four nations, China (11.0 [CI: 10.0, 11.8] thousand deaths) and the US (3.5 [CI: 3.2, 3.8] thousand deaths) have the next highest numbers of infant deaths following India.

By contrast, the production emissions in most of the G7 nations and Australia cause few infant deaths (Supplementary Fig. 2 in the SI), making the average age of death in such nations higher than the age in their footprint. Remarkably, Australia (7 years), Japan (6 years), and Canada (5 years) cause more than 5 years' difference in the average age of death between production emissions and footprint (Fig. 4). These disparities confirm that there is a clear heterogeneity between production and footprint not only in the number of deaths but also in the age distribution of mortality. In other words, a shift in focus to the consumption of these nations, rather than their production, creates the potential for avoiding substantial loss of the length of human life. Unpacking the structure of the footprint-based responsibility brings about a clear rationale for those nations to address this challenge (Fig. 5). In India, for instance, although domestic infant deaths (44.1 [CI: 40.2, 47.7] thousand) induced by the country's own consumption remain high, the US (1.3 [CI: 1.2, 1.4] thousand), the UK (0.37 [CI: 0.34, 0.40] thousand), Germany (0.35 [CI: 0.32, 0.38] thousand), Japan (0.26 [CI: 0.24, 0.29] thousand), and France (0.21 [CI: 0.19, 0.22] thousand) are responsible for a non-negligible number of infant deaths there.

As long as responsibility for infant deaths due to production emissions is the only issue pursued (Supplementary Fig. 3 in the SI), we can find no rationale for the G7 nations and Australia to confront the mass death of infants in India, China, Indonesia and South Africa except 10 deaths in China due to transboundary

pollution from Japan. Focusing solely on production emissions also creates a little opportunity for these developed countries to take responsibility for deaths outside the G20 nations, in particular in Africa and Asia. On the other hand, each footprint of the countries concerned highlights the involvement of infant deaths also in non-G20 countries in Africa and Asia. For instance, the US footprint involves substantial infant deaths in non-G20 countries (1.5 [CI: 1.4, 1.6] thousand), many of them in Africa (0.55 [CI: 0.50, 0.59] thousand), and Asia (0.66 [CI: 0.60, 0.71] thousand) (Fig. 5).

**Total premature deaths caused by G20 lifetime consumption.** Converting the G20's aggregate footprint to a per-capita footprint shows that individual consumption in the G20 nations leads on average to 0.46 [CI: 0.39, 0.53] × 10$^{-3}$ premature deaths per year (Supplementary Table 2 in the SI), with this impact continuing every year until death. Multiplying the per-capita footprint by the present life expectancy of G20 consumers permits a rough estimation of the total number of premature deaths due to individual lifetime consumption, which certainly differs across nations (Fig. 6). Taking the average life expectancy of G20 nations as 77.46 years, as in 2018, G20 per-capita lifetime consumption translates to 0.036 [CI: 0.030, 0.041] deaths; taking the reciprocal, the lifetime consumption of every 28 [CI: 24, 33] G20 citizens leads to the premature death of one person. For consumption in the G7, with a smaller per-capita footprint than the G20, the mortality figure for per-capita lifetime consumption is the same as for the G20: 0.036 [CI: 0.029, 0.043], because of about 4 years' longer life expectancy. Among the G20, China has the highest value: 0.051 [CI: 0.044, 0.058] deaths/capita, or 20 [CI: 17, 23] people's lifetime consumption costing one life, followed by Germany and the UK (both 0.044 [CI: 0.036, 0.052] deaths/capita), Russia (0.038 [CI: 0.031, 0.045] deaths/capita), and France (0.037 [CI: 0.030, 0.044] deaths/capita).

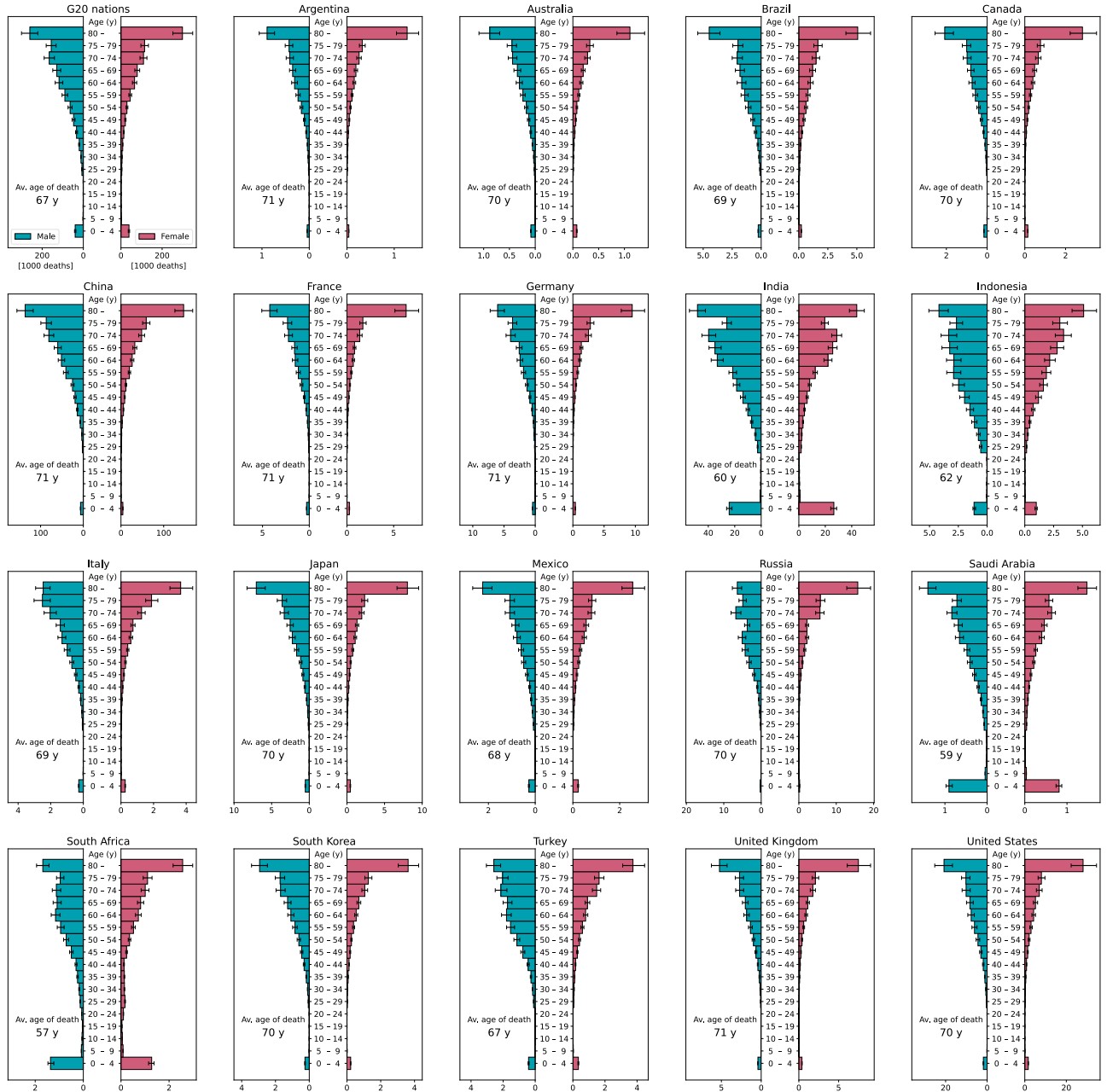

**Fig. 3 Age and gender composition of PM$_{2.5}$ premature death footprint in each G20 nation.** Error bars indicate 95% confidence intervals. The centre of the bars indicates the mean value.

When the average life expectancy exceeds 80 years, the difference in the footprint of each country tends to decrease, ranging from 0.021 [CI: 0.017, 0.026] in Australia to 0.044 [CI: 0.036, 0.052] in Germany. Furthermore, focusing on Japan (84.21 years), Italy (82.95 years), and Australia (82.75 years), the three longest-living countries in the world, the lifetime consumption footprints are found to converge at around 0.02–0.03 deaths/capita, implying that it is hard to decouple longevity and consumption-induced deaths. In addition, it is noteworthy that countries with a longer life expectancy (>80 years) account for the majority of foreign deaths in their footprint. This indicates that countries with longer life expectancies need to be take greater responsibility for the global impacts of their footprints.

South Africa has the shortest life expectancy among G20 nations (63.86 years), followed by India (69.72 years), but their lifetime-induced deaths are close to the figure for Japan, with the highest

longevity. For other G20 nations, with a life expectancy between 70 and 80 years, lifetime-induced deaths show a degree of spread. Importantly, it is hoped that the long-lived nations of the G20 will lead the way in technological, medical and economic cooperation, so the two aforementioned countries can follow the path to longer life expectancy without increasing their lifetime-induced deaths.

**Uncertainty in the footprints**. The present study employed the annual average PM$_{2.5}$ concentrations in 2010 compiled in GBD2016 (Global Burden of Disease)[36,37] to estimate the total premature deaths for each grid square. These deaths were then allocated to G20 nations based on their consumption-based PM$_{2.5}$ concentrations calculated by the regional chemical transport model CMAQ[38]. Global PM$_{2.5}$ concentrations based on ground measurements for each of 1.4 million grid squares in GBD are not available owing to the low density of PM$_{2.5}$ monitoring. Gridded

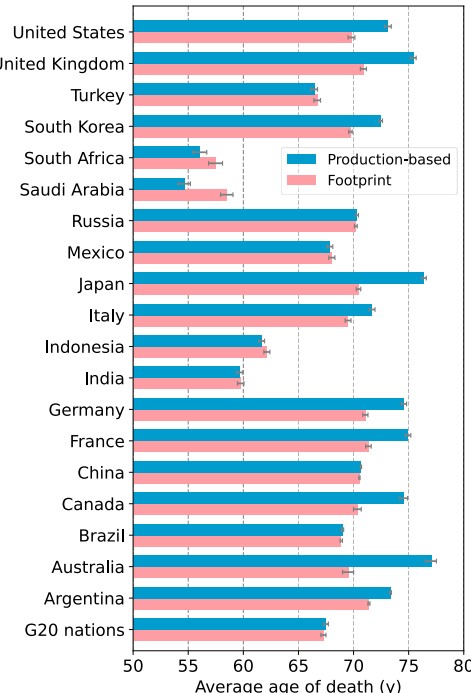

**Fig. 4 G20 comparison of average age of death for PM₂.₅ premature death footprint and that for premature deaths by production-based PM₂.₅ emissions.** Error bars indicate 95% confidence intervals. The centre of the bars indicates the mean value.

PM₂.₅ concentrations for all areas were therefore estimated using the data integration model with ground monitoring data, estimates of PM₂.₅ from remote-sensing satellites and chemical transport models within a Bayesian hierarchical modelling framework[37]. Hence, the 95% CIs of grid-square concentrations in GBD2016 are not necessarily narrow, as visible in the sometimes wide clusters of black dots in Supplementary Fig. 4 in the SI, which is a factor of uncertainty in the footprints. The uncertainty of the footprints with IER (mean) was evaluated in the context of extreme conditions of gridded PM₂.₅ concentrations in GBD2016 (gridded concentrations for each grid square are lower or upper bounds of 95% CI concentrations).

Taking the lower bound of the 95% CI of GBD2016, the footprints become 0.64 times (South Africa) to 0.81 times (China, India) lower, with an average of 0.72 times for the G20 nations (Fig. 7, left). With the upper bound concentrations, they increase between 1.11 times (India) and 1.24 times (South Africa), with a G20 average of 1.17 times. On the other hand, the footprint based on the 2010 average concentration in GBD2013[39] with the dust concentration data employed in this study gives a G20 mean of 0.90 times. The gridded concentrations directly calculated by the CMAQ give a G20 mean of 0.95 times, showing similar changes in the footprint as in the case of GBD2013. With both the direct CMAQ and GBD2013, the footprints of the G20 countries are generally within the range of the upper and lower bounds of the CI of GBD2016, which would support the reasonability of allocating total premature deaths in a grid based on GBD2016 concentration data by the consumption-based PM₂.₅ concentration computed by the CMAQ. In the case of Indonesia; however, the footprint with the CMAQ (1.39 times) exceeds the upper bound case. Even at the upper bound of the CI, there is only one grid square with a PM₂.₅ concentration over 30 μg/m³ among Indonesian grid squares with a population density of 22.5 persons/km² or more (Supplementary Fig. 4 in the SI), despite some studies observing concentrations as high as 40–50 μg/m³ in

2001–2007[40], 33 μg/m³ in 2014[41], and 27–69 μg/m³ in 2007[42] in Bandung City. This suggests a possible underestimation of Indonesia's footprint.

The CMAQ-simulated concentration used for the premature death allocation is impacted by the accuracy of the air pollutant emissions inventory. Uncertainty has not been estimated for the emission inventory of EDGAR v4.3.1[43] used in this study. However, for the updated EDGAR v4.3.2, uncertainty estimates are provided for air pollutant emissions, based on the estimated uncertainty of the activity data and emission factors for each emission sector, pollutant and country[44]. The two EDGAR databases are consistent for almost all sources and there is no significant discrepancy in global emissions between them except for NMVOC[36]. The uncertainties in the EDGAR v4.3.2 emission inventory can therefore be taken as holding for EDGAR v4.3.1, too. The uncertainties for primary PM₂.₅ vary by region, with estimates ranging from 49.4 to 96.8% in 2010, as do those for precursors of secondary particles: of these, NH₃ (185.0–294.6%) has the highest uncertainty, while SOₓ (12.5–48.3%) and NOₓ (17.9–117.9%) are reported to have relatively small estimation errors[44].

When inputting the upper and lower bound emissions for 95% CI of EDGAR v4.3.1 to CMAQ, the computed PM₂.₅ concentrations in any grid square are approximately proportional to those with average emissions (Supplementary Fig. 5 in the SI). There are infinite combinations of possible emissions, not merely with all grid squares taking upper or lower bounds at the same time. However, in so far as the overall emissions in the inventory data tend to be either underestimated or overestimated compared with the mean emissions, the impact on the premature death footprint allocated by the CMAQ concentrations would be modest, as concentrations vary proportionally to emissions. Within many territories, a 45° line (CMAQ concentration with average emissions) is generally located between the concentrations with the upper and lower emissions. As long as the emissions vary randomly and uniformly between the upper and lower bounds, the average concentration value is expected to approach the concentration with the average emissions. Taking the concentration with the average emissions, as has been done in this study, is therefore considered a good approximation of the footprint based on the representative allocation rate.

Although the CMAQ has a spatial resolution of 45-km grid squares for more precise country-by-country analysis, at the expense of detail the world was divided into six domains to reduce the computational burden. This division makes it impossible to analyse the impact of consumption-induced concentrations outside each domain, although, for instance, Zhang et al.[12] estimated that 2.2 and 2.0% of premature deaths within the US and Canada, respectively, in 2007 were attributable to long-range transboundary pollution from China. We therefore quantified the contribution of G20 consumption-induced concentrations to the concentrations in the grid squares located on the eastern, western, northern and southern edges of each domain (Supplementary Figs. 6–11 in the SI). For example, at the four edges of the Asian region, the largest contribution of China's consumption to the PM₂.₅ concentration was 26%, in a grid square with about 5 μg/m³. Across all peripheral grid squares, the average concentration contributed by China's consumption was 0.14 μg/m³, suggesting that while the impact outside the region is non-zero, it is sufficiently small compared with that inside the region.

The 95% confidence intervals for the number of premature deaths reported above reflect the uncertainty of the IER (Integrated Exposure-Response) model[32,45] that determines the relative risk associated with ambient PM₂.₅ exposure. The IER function has been used in many previous footprint studies[12,17,21,23]. Recently, the developers of the IER function

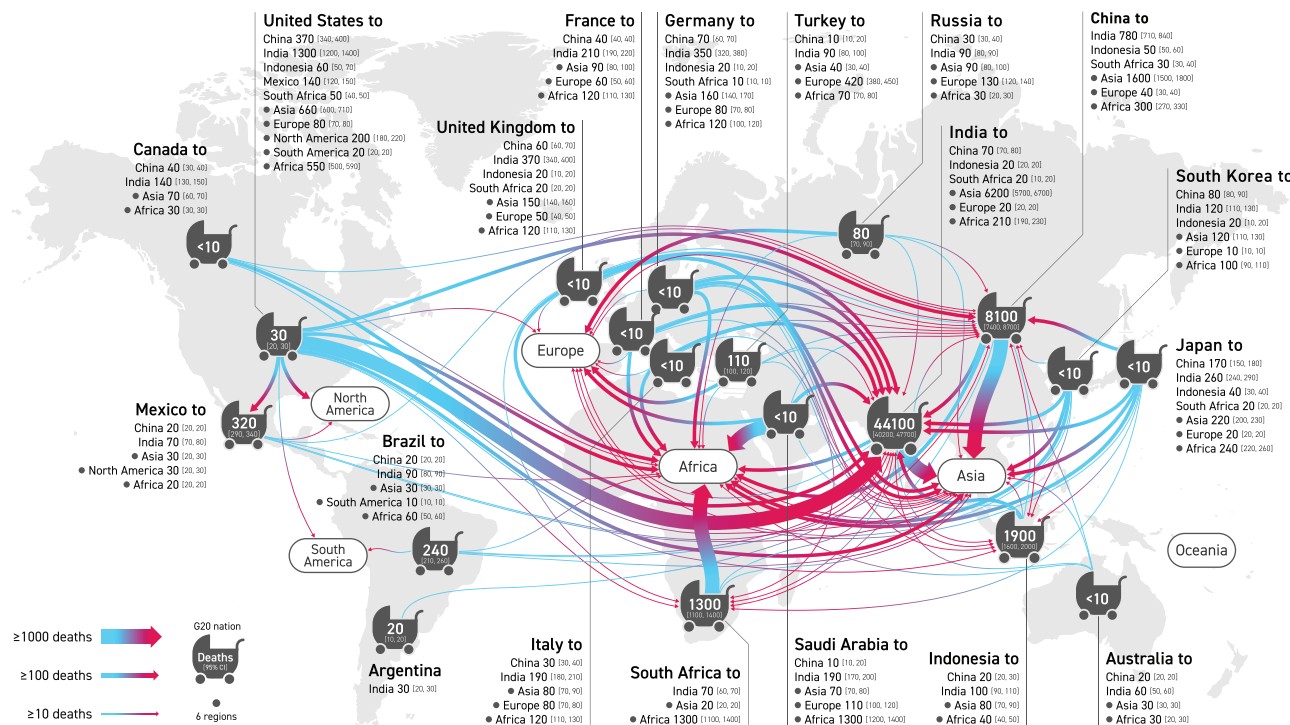

**Fig. 5** Relationships between consumer countries and impacted countries ("to") for infant deaths (zero to less than five years old) in PM$_{2.5}$ premature death footprint of G20 nations.

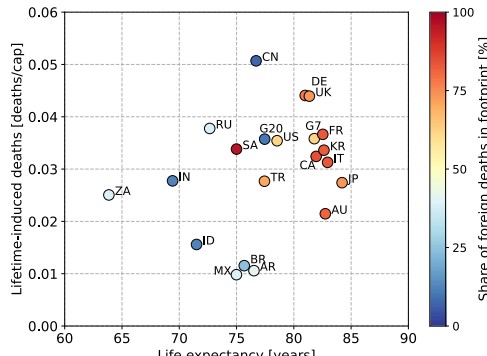

**Fig. 6 Relationship between life expectancy, PM$_{2.5}$-related premature deaths induced by lifetime consumption per capita and percentage of foreign deaths in PM$_{2.5}$ premature death footprint in each G20 nation.** Country codes as follows: Argentina (AR), Australia (AU), Brazil (BR), Canada (CA), China (CN), France (FR), Germany (DE), India (IN), Indonesia (ID), Italy (IT), Japan (JP), Mexico (MX), Russia (RU), Saudi Arabia (SA), South Africa (ZA), South Korea (KR), Turkey (TR), United Kingdom (UK) and United States (US).

themselves proposed the GEMM (Global Exposure Mortality Model)[46], which improves the relative risk estimation based on cohort studies of ambient PM$_{2.5}$ mortality, covering higher outdoor PM$_{2.5}$ concentrations, which has been a challenge in the IER model. Here, we estimated the premature deaths due to G20 footprints and those of production emissions using the GEMM 5-COD model and in each case compared them with the IER results in terms of the five diseases studied and exposures above 25 years' age targeted by the GEMM. Although there are differences among nations, the GEMM-based footprints are all 1.5–2 times larger than the IER footprints (Fig. 7 right). The extent of this increase is consistent with Burnett et al.[46], who confirmed a change in global premature deaths in 2015 from

4.002 million premature deaths based on the IER to 6.889 million deaths with the GEMM 5-COD. As for premature deaths within China, Liu et al.[47] report an increase from 0.986 million deaths with the IER to 1.681 million deaths with the GEMM 5-COD in 2010. In all nations except Australia, premature deaths due to production emissions showed a higher increase than for the footprint, confirming the trend that uncertainty in footprint deaths is less affected by the choice of the relative risk model.

## Discussion

In this study, we found that consumption in the 19 nations with presidency rights to the G20, which can set the agenda for the G20 summit, induces the mortality of 1.983 [CI: 1.685, 2.285] million people a year through global supply chains as a result of PM$_{2.5}$. Our analysis suggests that the G20 meeting will be further enriched as a forum for decision-makers to discuss joint international measures to curb PM$_{2.5}$-related premature deaths by extending the interrelationships among nations to include these consumption-based footprints.

In this respect, the G20 offers greater potential than similar high-level policy meetings like the G7 (Canada, France, Germany, Italy, Japan, UK, US), as consumption by the G7 nations induces far fewer premature deaths worldwide: 323 [CI: 263, 386] thousand. In addition, 62% of these deaths occur outside the G7, making it difficult for PM$_{2.5}$ control to become a joint issue for the G7, while in the G20's footprint, only 11% of deaths are in non-G20 nations. One point the G7 and G20 have in common is that about half the total premature deaths are attributable to secondary particles, confirming the importance of mitigation measures for secondary particle precursors, including VOCs and ammonia, as well as for fuel combustion, although these are generally identical to measures addressing primary PM$_{2.5}$.

Among the 19 G20 nations, the consumption of eleven nations induces over 50% of premature deaths in other countries. Strikingly, many of the bilateral relationships involved in each nation's

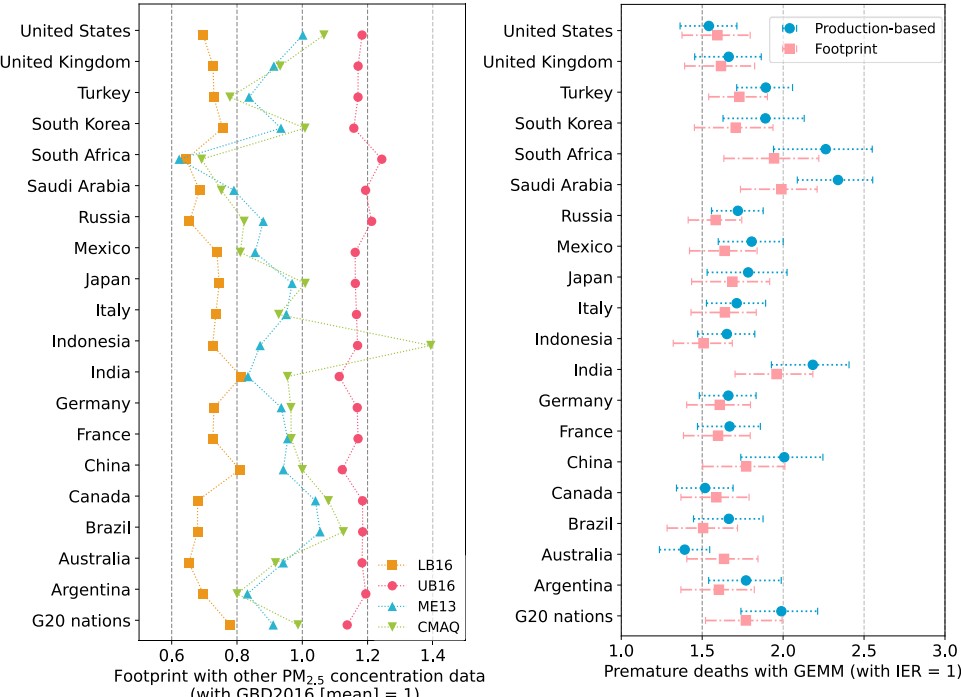

**Fig. 7 (Left) Change in premature deaths footprint when the other PM₂.₅ concentration data (LB16, UB16, ME13, CMAQ) are used for grid-square concentrations to estimate total premature deaths in each square, relative to 1 for premature deaths with the IER (Integrated Exposure-Response) model (50% value) and PM₂.₅ concentration (mean) in GBD2016 (Global Burden of Disease 2016). (Right) Change in number of premature deaths (age 25 years and older) for footprint and production-based emissions when GEMM (Global Exposure Mortality Model) is used for relative risk calculation, relative to 1 for premature deaths with IER model (50% value). Error bars indicate 95% confidence intervals (CI).** LB16 and UB16 are the cases with the lower and upper bound concentration in 95% CI of GBD2016, respectively. ME13 is the case with GBD2013 concentration (mean) in 2010 and CMAQ is the case with concentration calculated by the CMAQ (Community Multiscale Air Quality) modelling system. The cases of IER with upper and lower bounds of the 95% CI are shown in Supplementary Fig. 12 in the Supporting Information.

footprint connect to China and India, which have the largest absolute number of premature deaths, a fact that is overlooked if the focus is solely on production-based emissions. The pursuit of consumer responsibility for PM₂.₅ in the G20 will thus pave the way to reducing premature deaths in these two countries. There is one G20 nation, however, that does not feature among the top 10 countries with the most premature deaths occurring in the footprint of each of the 19 nations: South Africa. Unlike in the case of China and India, joint action to reduce mortality in South Africa would not appear to be an obvious priority if only general consumer responsibility is pursued. Like India and China, South Africa is a country with a high incidence of infant mortality due to PM₂.₅. If South Africa were to be omitted from the framework of joint measures, it would only be in this country that premature infant deaths remained unchanged. One approach to including South Africa would be to clarify consumer responsibility by focusing on the infant victims. The visualized network for PM₂.₅ premature infant deaths (Fig. 5) reveals the opportunities for involvement by the US, the UK, Japan, China, India and Germany in the protection of South African infants. Infant mortality hampers achievement not only of Goal 3 (Good health and well-being) of the Sustainable Development Goals (SDGs)[48] but also that of Goal 4 (Quality education), by depriving children of access to education. There is no doubt that the achievement of Goal 17 (Partnerships for the goals), backed by footprint-based responsibility and rationale, will be key to overcoming this challenge.

It should be noted that the effectiveness of measures focusing on mitigation of premature infant death will be limited unless the basic infant mortality rate is improved in conjunction with decreases in PM₂.₅ concentrations. For example, the LRI mortality rate among male infants is low in France (0.01 deaths per 1000 people) and

Germany (0.02 deaths per 1000), but high in India (1.97 deaths per 1000) and South Africa (1.92 deaths per 1000), representing an almost 200-fold gap[49]. If this gap is not reduced, infant mortality will remain concentrated in certain countries. Especially in Asia and Africa, future population growth is predicted, which implies that delays in abatement measures would permit an unnecessary increase in premature infant deaths in those regions.

An additional concern is that the number of premature infant deaths due to PM₂.₅ may potentially be even higher than the figure reported in this study. Footprints using the GEMM, the latest relative risk model, showed a 1.5- to 2-fold increase in the number of deaths in the 25+ age group compared with the footprints calculated with the IER. However, the current GEMM model is unapplicable for evaluating the under-25 age group, including infant mortality, and it is quite possible that the number of infant deaths will increase in the same way as for other age groups. This concern calls for urgent development of a relative risk model to reduce the uncertainties in PM₂.₅-driven mortality, which will serve to improve the accuracy of estimates of comprehensive PM₂.₅ health damage and economic impact[50–53], as well as approximating the true number of infant victims.

At the upper and lower bounds of uncertainty in the PM₂.₅ concentration data, the G20 country footprints vary by an average of 0.72 and 1.17 times, respectively, which seems less severe than the impact of the choice of relative risk model described above. To significantly reduce the uncertainty of the PM₂.₅ concentration data would require global expansion of ground monitoring points for PM₂.₅ concentrations using universally standard measurement methods, but this is prohibited by both time and cost constraints. There is therefore little likelihood of any major decrease in the uncertainty of the footprints in the near term. The critical actions

by policymakers called for here are that the G20 nations agree on a unique choice of footprint within the given range of uncertainty and that the G20 takes joint action as soon as possible to reduce the number of premature deaths due to its consumption. If these nations agreed on their smallest footprint within the uncertainty, it would be 0.78 times the mean case, which excludes 439 thousand deaths from the mean G20 footprint. Yet, long-term postponement of reduction action means G20 consumption will continue to result in more premature deaths than the number of deaths excluded. We believe the G20's responsibility for its footprint includes the responsibility to choose and agree on a single figure without delay, within the limits of its uncertainty.

## Methods

**Four stages in an interdisciplinary approach**. This study combines the findings of research conducted using interdisciplinary models and undertaken in four broad stages. Stage 1 focused on the creation of a footprint emissions map and production-based emissions map for each of the 19 nations eligible for the G20 summit presidency. Stage 2 was concerned with calculating the respective ambient $PM_{2.5}$ concentrations deriving from the two maps. Stage 3 estimated the health impacts due to exposure to the $PM_{2.5}$ concentrations calculated in Stage 2 in the 199 impacted countries that are ultimately affected by consumption in the G20. The health impacts in the countries affected by the production-based emissions of each G20 nation were also computed. Stage 4 involved estimating the total premature deaths induced by lifetime consumption per capita, considering the average life expectancy of each G20 nation.

**Stage 1: consumption-based emissions map**. In Stage 1, the global multiregional input-output (MRIO) model Eora[54] for the year 2010, which incorporates global supply chains among 187 countries, and the emissions inventory map EDGAR v.4.3.1[43] were combined using the spatial footprint analysis [55,56], which enabled the mapping of emissions by sectors and by countries determined by an environmental input-output analysis[17].

By inputting the domestic final consumption of each G20 nation to Eora, we extracted only emissions globally induced by the consumption from the EDGAR emissions map. *In concreto*, a footprint emissions map in 2010, or consumption-based emissions map, of $PM_{2.5}$ primary particles (black carbon, organic carbon and other primary components) and secondary precursors ($NO_x$, $SO_2$, $NH_3$, CO, and NMVOC) was prepared for each nation. The production-based emissions map, showing the direct emissions generated in each G20 nation, was also compiled from the EDGAR emissions map by extracting the emissions within the territory of each nation.

**Stage 2: ambient $PM_{2.5}$ concentration**. In Stage 2, consumption-based $PM_{2.5}$ concentrations for each G20 nation were calculated using the above consumption-based emissions map and atmospheric simulation models. Zhang et al.[12] demonstrated that it is more significant for $PM_{2.5}$ consumption-based accounting to understand the effects of induced $PM_{2.5}$ emissions in other countries through international trade than the health impacts of long-range transboundary pollution. Therefore, aiming to more accurately capture the impact associated with the emitting countries in finer resolution, this study used the regional chemical transport model CMAQ (version 5.2.1)[38] as the air quality model and the regional meteorological model WRF (version 3.8.1)[57], and set the grid size to 0.5° × 0.5° (about 45 km × 45 km) squares. The vertical airspace was divided into 25 layers extending to 100 hPa. The global chemical transport model MOZART-4[58] was used to calculate concentrations at the boundaries of the calculated region. As the detailed grid size increases the cost of computation enormously, we divided the world into six regions: Asia, Europe, North America, South America, Africa and Oceania.

The models computed hourly $PM_{2.5}$ concentrations (over 12 months from January 1 to December 31, 2010) in each grid square using the original emissions map of EDGAR (v.4.3.1) and other emissions map data (GFED (Global Fire Emissions Database) v4.1s[59] and MEGAN (Model of Emissions of Gases and Aerosols from Nature) v2.1[60]) (below: 'base-case emissions'). The annual average of hourly concentrations (below: 'base-case concentrations') was defined as the $PM_{2.5}$ concentration for that grid square. The base-case concentrations in the G20 nations were compared with the GBD2016[36,37] data with 95% CI, while concentrations in countries other than the G20 nations were summarized for each of the six regions as a whole, owing to the limited data available for comparing calculated and measured secondary particle concentrations, and compared with the observation data in IMPROVE[61], EMEP[62], and EANET[63] (Supplementary Fig. 4 in the SI).

The annual $PM_{2.5}$ concentrations simulated by CMAQ generally fall into the 95% CI of the GBD2016 data. Most of the high GBD2016 $PM_{2.5}$ concentrations deviating furthest from the simulated values are affected by dust, which CMAQ has difficulty simulating accurately. Influences of dust were corrected in the following Stage 3. While no grid squares are affected by dust in Argentina, Mexico or Turkey (Supplementary Fig. 4 in the SI) according to the GBD2013 criteria ($PM_{2.5} \geq 36\ \mu g/m^3$ and dust fraction ≥50%), in those countries, $PM_{2.5}$ may still, in

fact, be affected by dust to a certain extent, as has indeed been reported[64–68]. In developing the GBD2016 data, ground monitoring data, remote-sensing satellite data and chemical transport models were integrated and good correlations with the GBD2016 data therefore do not necessarily mean better model performance for ambient $PM_{2.5}$ concentrations. Supplementary Fig. 4 in the SI reveals narrower variations in the $PM_{2.5}$ concentrations from the GBD2016 data than the values simulated by CMAQ. However, such features are not found in comparisons of the observed[69] and CMAQ-simulated AOD (Aerosol Optical Depth) (Supplementary Fig. 13 in the SI), suggesting that variations in actual ambient $PM_{2.5}$ concentrations are greater than in the GBD2016 data and were reproduced well by CMAQ. In particular, large deviations in Indonesia suggest possible underestimations in the GBD2016 data. Exceptional positive biases of AOD in Argentina may be due largely to biomass-burning emissions and transport from the Amazon, where there were intense forest fires in 2010, to northern Argentina[70]. Correlations in $SO_4^{2-}$, $NO_3^-$, and $NH_4^+$ concentrations indicate good performance of CMAQ on secondary components of $PM_{2.5}$.

$PM_{2.5}$ concentrations, excluding the contribution of each nation's consumption-based emissions, were then computed using the emissions map, with the nation's consumption-based emissions being subtracted from the base-case emissions map. This was done in the same way described above, calculating the hourly values and then determining the annual average. Then, by subtracting the concentrations thus obtained from the base-case concentrations, the consumption-based $PM_{2.5}$ concentrations of the nation were determined (Supplementary Figs. 14–19 in the SI). The $PM_{2.5}$ concentrations from the production-based emission maps were computed using the same models and calculation conditions. Therefore, the production-based emissions are linked to only production activities, resulting in no inclusion of dust and forest fires emissions. This causes discrepancy from the general particle compositions within each country's territory.

**Stage 3: health impacts due to ambient $PM_{2.5}$**. In Stage 3, premature deaths in each age group were estimated in each grid square. Population distribution data by sex and age[71] were prepared for each 0.1° × 0.1° grid square, and to define the consumption-based concentrations to this level of precision, the CMAQ-based consumption-based concentrations of a nation from Stage 2 were adjusted with the average $PM_{2.5}$ concentrations in 2010, based on GBD2016[37], within each 0.1° × 0.1° grid square. Specifically, the consumption-based concentration of the nation at grid resolution was determined by multiplying the average GBD2016-based $PM_{2.5}$ concentrations in each 0.1° × 0.1° grid square by the ratio of the nation's CMAQ-based consumption-based concentration to the base-case concentrations on a 0.5° × 0.5° grid, where the GBD and CMAQ grids overlap. To improve the accuracy of the dust concentrations of $PM_{2.5}$ in each grid square, the dust concentrations in 2010 provided by GBD2013[39] were extrapolated to the respective corresponding 0.1° × 0.1° grid squares. When the dust concentration of a grid square exceeded the total $PM_{2.5}$ concentration of the grid in GBD2016, the dust concentration was replaced by the total concentration.

Using the IER model[32,45] with the parameters reported in Supplementary Table 2 in the SI, the relative risks (50, 97.5, and 2.5% values) of five diseases (LRI, COPD, LC, stroke, and IHD) (Supplementary Fig. 20 in the SI) were calculated by age (five age groups from 0 to over 80 years old) and the number of premature deaths in each 0.1° × 0.1° grid square under base-case concentrations were determined using the same method as Apte et al.[32]. Then, the number of premature deaths due to consumption-based concentrations was determined using the Direct proportion of burden method[12,23], which assumes that the relative risk is proportional to the concentration induced by the country concerned. *In concreto*, the total number of premature deaths in each 0.1° × 0.1° grid square was allocated to G20 nations based on the share of their consumption-based concentrations in the GBD-based concentrations in that square. The same procedure as above was applied to estimate the premature deaths associated with the $PM_{2.5}$ concentrations originating from production-based emissions.

**Stage 4: premature deaths induced by lifetime consumption**. In Stage 4, a premature death footprint per capita in 2010 was calculated by dividing the premature deaths induced by consumption in each G20 nation obtained above by the population of each nation (Supplementary Table 3 in the SI). Finally, lifetime-induced deaths per capita were estimated by multiplying the premature death footprint per capita by the average life expectancy in each G20 nation.

## Data availability

The sources of all the data used in this study are duly referenced. All relevant data are available from the corresponding author upon reasonable request. All the data on consumption (footprint)- and production-based premature deaths and infant deaths for the 19 G20 nations based on the two cases—with the IER model[32,45] and GBD2016[36,37] and with the GEMM[46] and GBD2016—are provided as a Source Data file provided with this paper. Source data are provided with this paper.

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

## Acknowledgements

This research was supported in part by a Grant-in-Aid for Research (No. 16H01797) from the Japanese Ministry of Education, Culture, Sports, Science and Technology, and by a Fund for the Promotion of Joint International Research (Fostering Joint International Research (A)) of the Japan Society for the Promotion of Science (No. 18KK0322). We are also grateful to Nigel Harle of Gronsveld, the Netherlands, for his conscientious improvement of our English.

## Author contributions

K.N., S.T., S.C., K.K., S.K., and Y.K. designed the study; K.N., S.T., S.C., and K.K. performed the analyses; and K.N., S.T., S.C., K.K., W.T., and M.L. prepared the manuscript.

## Competing interests

The authors declare no competing interests.
