## [Peer Review File · Nature Communications]

REVIEWER COMMENTS

Reviewer #1 (Remarks to the Author):

The manuscript by Nansai et al. presents valuable new analysis of the human health impacts of consumerism, quantifying how many premature deaths are caused by emissions associated with international consumption in G20 nations. The work goes beyond previous efforts in terms of the treatment of air quality simulations and the scope of countries considered. Still, in flavor it is an incremental step beyond e.g. Zhang 2017, which essentially looked at the same sort of analysis yet with 13 regions rather than G20 countries. Zhang 2017 also used a coarser air pollution model, but to their benefit over the current study they included long-range atmospheric transport and verified the accuracy of their base case simulation using comparison to observations, neither of which are included nor discussed in the present work. In total, it doesn't not strike me as a major technical advancement over prior work, and the writing needs to be more modest in this regards. However, I think in terms of their application and analysis the work is still suitable for Nat Com if this point and others below are addressed in a revised manuscript.

- The background on global PM2.5 health impacts could be more comprehensive; the authors reference WHO estimates but there are others, some of which range up to 8 million premature deaths per year from long-term PM2.5 exposure (Burnett et al., PNAS, 2018). Overall, there are significant uncertainties in these sorts of estimates, as evident in their range, which should be recognized.

- There isn't any direct comparison of the results presented here to previous work, even for the subset of results that are directly comparable. For example, on line 124 - 125: Some of these directly comparable to results in Zhang 2017. Here the impact of US consumption on deaths in the US and China are 54k and 38k, respectively, whereas in Zhang 2017 it is and 62k and 54k. The relative amounts seem similar, but the absolute values in the present study are lower. Please add discussion explanation and also look for other places where specific results or broader conclusions can be compared to prior work.

- Atmospheric transport of PM associated with consumption-related emissions for specific countries across their 5 global regions is neglected. This neglect isn't mentioned nor considered, but it should be. From Zhang 2017, for example China's emissions impacted 2.2% of the deaths in the US (1.5k). In total, China's emissions impact 5,600 deaths that lie outside the Asian domain considered here.

- I found the discussion on cause and gender distracting and less interesting / tangential. The main story is about international inequities for total health impacts. The study would be stronger to remove this section and instead add material in response to other lackings mentioned in this review.

- the lifetime projections are a really neat finding, and will make very noteworthy news splashes / press releases. However, they seem flawed and subject to misinterpretation. Allowing only population to grow but not accounting for changes in mortality rates presents an inconsistency. Growing the footprints as a linear scaling with population also seems unfounded. Over a 70 yr lifetime, emissions regulations are going to change the emissions so dramatically that such an assumption seems absurd. Also the spatial distribution would change, and atmospheric transport will change (owing to GHG-driven climate change). In total, the authors have addressed only a small fraction of the components needed to make accurate forecasts, so I recommend not making such forecasts at all, as the nuances of what is included and what is neglected will be overlooked in summaries. Instead, just use the present day conditions extended over the lifetime, and present results with the phrase: If current pollution, population, and consumption levels remained constant, the lifetime consumption in country X would claim Y lives. This number is much easier to understand.

Throughout — there is little to no discussion of uncertainty. This needs to be addressed and quantified along the following 3 parts of the analysis: emissions estimates, atmospheric modeling, health impact responses. Some of these are going to have uncertainties of at least x2 — thus it seems misleading to present other results to 4 significant figures. Comparison of model estimated PM_{2.5} concentrations to measurements (in situ or even satellite derived) is required. Comparison of total burden of disease in 2010 with the base case simulation to prior studies is required.

The sudden introduction of GHG implications is unsupported and not well considered. GHG emissions do not well correlate with PM_{2.5} health impacts, because it is possible to have high-carbon low-pollution economies (like the US). See for example Anenberg et al, Sci Reports, 2020. Please just remove this discussion altogether.

308-311: This is an odd conclusion — how is one going to have policy that counters population growth and aging? Regardless, just remove the component of the analysis owing to population projections, as recommended above.

Reviewer #2 (Remarks to the Author):

This study combined an interdisciplinary model to study the average annual environmental PM2.5 concentration based on consumption in the 19 nations eligible for the G20 summit presidency and calculated the number of premature deaths in each country. The 2010 per capita lifetime consumption estimates the number of premature deaths from 2050 to 2100. While I think this study has some merits with significantly more work, it, at present, misses some key information to have confidence in the results.

First, and foremost, there is no validation on the model results. To be publishable, they need to show that the model adequately captures the concentration levels for PM2.5, especially for the fractions of primary and secondary PM2.5 they emphasized in the manuscript. In addition, there are published results about the premature mortality associated with exposure to outdoor PM2.5 for countries they focused on here (for examples, Lelieveld et al., *Nature* 2015, 525 (7569), 367–371; Liu et al., *Sci. Total Environ.* 2016, 568, 1253–1262; Hu et al., *Environ. Sci. Technol.* 2017, 51, 9950–9959). They should also compare their estimates in the basecase to those published results to build confidence.

Second, it is unclear how they attributed the premature mortality to primary vs. secondary. Did they apply the fractions of primary and secondary in PM2.5 concentrations to the total premature mortality? Or they estimated the premature mortality using the primary and secondary concentrations? In any case, they assumed the same exposure-response functions for the primary and secondary PM2.5. Although it is a 'standard' treatment, the novelty of the manuscript emphasizing the 'secondary' fraction in the results is not very convincing.

Third, it is a little bit outdated to use 2010 as the basecase year. In addition, the fourth stage of the method is to calculate the per capita lifetime cost based on the data from January 1 to December 31, 2010, to estimate the total premature deaths in 2050 and 2100. One-year data is used to predict the long-term future impact based on population size changes alone. It is not convincing and has too many uncertain factors to make research meaningful.

Minor comments:

1. Where does the global supply chain data come from, and what data is needed for input in the global multi-regional input-output (MRIO) model?

2. The method part lacks model configuration, model verification, and this part needs to be supplemented, such as the MRIO model, CMAQ model and weather research and forecast model
3. How the territorial emission maps are created is not clear.
4. The number for India in Fig. 6 is 44126?
5. Are all the numbers reported in the manuscript the median estimates? To be more meaningful, 95% Cis should be included and some discussions should be provided about the uncertainties in the estimates.

Responses to Comments by Reviewer 1

Comment 1

The manuscript by Nansai et al. presents valuable new analysis of the human health impacts of consumerism, quantifying how many premature deaths are caused by emissions associated with international consumption in G20 nations. The work goes beyond previous efforts in terms of the treatment of air quality simulations and the scope of countries considered. Still, in flavor it is an incremental step beyond e.g. Zhang 2017, which essentially looked at the same sort of analysis yet with 13 regions rather than G20 countries. Zhang 2017 also used a coarser air pollution model, but to their benefit over the current study they included long-range atmospheric transport and verified the accuracy of their base case simulation using comparison to observations, neither of which are included nor discussed in the present work. In total, it doesn't not strike me as a major technical advancement over prior work, and the writing needs to be more modest in this regards. However, I think in terms of their application and analysis the work is still suitable for Nat Com if this point and others below are addressed in a revised manuscript.

Response

We appreciate your constructive comments and your seeing the value of expanding the previous study by Zhang et al. (2017) to nation-by-nation consumer responsibility, focusing on the countries of the G20, an entity with a specifically international high-level decision-making process.

The paper has been revised, focusing on the analysis of uncertainty. Specifically, (1) 95% confidence intervals (CI) for calculated values are now reported, (2) the difference in key mortality figures between our study and the previous studies is explicitly stated, and (3) the concentrations simulated by the air quality model CMAQ are compared with GBD-reported values. In addition, regarding the concern about the effect of long-range atmospheric transport on mortality, which our model is unable to handle, we have now quantified the contribution of G20 consumption to the PM_{2.5} concentration in the grids squares on the edges of each calculation domain.

We have also removed results on the disease-specific analysis and added an uncertainty section to explain the impact of using the GEMM on our PM_{2.5} footprint analysis. The discussion section has also been revised to reflect these changes.

Comment 2

The background on global PM_{2.5} health impacts could be more comprehensive; the authors reference WHO estimates but there are others, some of which range up to 8 million premature deaths per year from long-term PM_{2.5} exposure (Burnett et al., PNAS, 2018). Overall, there are significant uncertainties in these sorts of estimates, as evident in their range, which should be recognized.

Response

Thank you for your good suggestion. We have calculated the footprint-based and production-based premature deaths for each G20 country by applying the new relative risk assessment model (GEMM) developed by Burnett et al., PNAS, 2018 that you introduced. However, the calculation includes only five diseases (GEMM-5COD) and deaths over the age of 25 that could be compared with the results with the IER model. As shown in Figure 7, we have clarified that the application of GEMM increases the number of deaths in each footprint by a factor of 1.5 to 2 compared with IER, with this increase being higher for the deaths due to production-based emissions. The following sentences have been added to the main text.

Lines 256 - 273

The 95% confidence intervals for the number of premature deaths reported above reflect the uncertainty of the IER (Integrated Exposure-Response) model^{28,32} that determines the relative risk associated with ambient PM_{2.5} exposure. The IER function has been used in many previous footprint studies^{8,13,17,19}. Recently, the developers of the IER function themselves proposed the GEMM (Global Exposure Mortality Model)³³, which improves the relative risk estimation based on cohort studies of ambient PM_{2.5} mortality, covering higher outdoor PM_{2.5} concentrations, which has been a challenge in the IER model. Here, we estimated the premature deaths due to G20 footprints and those of production emissions using the GEMM 5-COD model and in each case compared them with the IER results in terms of the five diseases studied and exposures above 25 years' age targeted by the GEMM. Although there are differences among nations, the GEMM-based footprints are all 1.5 to 2 times larger than the IER footprints (Figure 7). The extent of this increase is consistent with Burnett et al.³³, who confirmed a change in global premature deaths in 2015 from 4.002 million premature deaths based on the IER to 6.889 million deaths with the GEMM 5-COD. As for premature deaths within China, Liu et al.³⁴ report an increase from 0.986 million deaths with the IER to 1.681 million deaths with the GEMM 5-COD in 2010. In all nations except Australia, premature deaths due to production emissions showed a higher increase than for the footprint, confirming the trend that uncertainty in footprint deaths is less affected by the choice of relative risk model.

Figure 7
Change in the number of premature deaths (aged 25 years and older) for footprint and production-based emissions when the GEMM (Global Exposure Mortality Model) is used for relative risk calculation, compared with 1 for premature deaths with the IER (Integrated Exposure-Response) model (50% value). Error bars indicate 95% confidence intervals.

Comment 3

There isn't any direct comparison of the results presented here to previous work, even for the subset of results that are directly comparable. For example, on line 124 - 125: Some of these directly comparable to results in Zhang 2017. Here the impact of US consumption on deaths in the US and China are 54k and 38k, respectively, whereas in Zhang 2017 it is and 62k and 54k. The relative amounts seem similar, but the absolute values in the present study are lower. Please add discussion explanation and also look for other places where specific results or broader conclusions can be compared to prior work.

Response

Comparison of premature footprint deaths could only be made with Zhang et al. (2017). However, it was hard to determine which factor caused the difference in the number of deaths, because the difference in mortality between Zhang et al. and our study was related to many factors, including emission inventories, MRIO model, air-quality models and the diseases and ages section considered, and in addition the years analysed: 2007 versus 2010. We appreciate your understanding of this limitation. On the other hand, several previous studies were available for comparing premature deaths due to production-based emissions. We therefore compared premature deaths worldwide, and for China and India, which have the largest numbers of deaths. The papers referred to are as follows:

- | | |
|----|---|
| 8 | Zhang, Q. et al. Transboundary health impacts of transported global air pollution and international trade. Nature 543, 705-709, doi:10.1038/nature21712 (2017). |
| 27 | Lelieveld, J., Evans, J. S., Fnais, M., Giannadaki, D. & Pozzer, A. The contribution of outdoor air pollution sources to premature mortality on a global scale. Nature 525, 367-371, doi:10.1038/nature15371 (2015). |
| 28 | Apte, J. S., Marshall, J. D., Cohen, A. J. & Brauer, M. Addressing Global Mortality from Ambient PM _{2.5} . Environmental Science & Technology 49, 8057-8066, doi:10.1021/acs.est.5b01236 (2015). |
| 29 | Liu, J., Han, Y., Tang, X., Zhu, J. & Zhu, T. Estimating adult mortality attributable to PM _{2.5} exposure in China with assimilated PM _{2.5} concentrations based on a ground monitoring network. Sci Total Environ 568, 1253-1262, doi:10.1016/j.scitotenv.2016.05.165 (2016). |
| 30 | Hu, J. et al. Premature Mortality Attributable to Particulate Matter in China: Source Contributions and Responses to Reductions. Environ Sci Technol 51, 9950-9959, doi:10.1021/acs.est.7b03193 (2017). |
| 31 | Conibear, L., Butt, E. W., Knote, C., Arnold, S. R. & Spracklen, D. V. Stringent Emission Control Policies Can Provide Large Improvements in Air Quality and Public Health in India. Geohealth 2, 196-211, doi:10.1029/2018GH000139 (2018). |

Comment 4

Atmospheric transport of PM associated with consumption-related emissions for specific countries across their 5 global regions is neglected. This neglect isn't mentioned nor considered, but it should be. From Zhang 2017, for example China's emissions impacted 2.2% of the deaths in the US (1.5k). In total, China's emissions impact 5,600 deaths that lie outside the Asian domain considered here.

Response

Thank you for this very significant point. Since Zhang et al. (2017) concluded that we should focus on PM_{2.5}-driven deaths in other countries induced through international trade rather than deaths by conventional transboundary pollution, in order to accurately measure the health impact near emission sources, the CMAQ with a 45km-by-45km grid was employed to calculate the PM_{2.5} concentration. However, at the same time, owing to the limitation of our computational load, we

divided the world into six domains (Asia, Europe, North America, South America, Africa and Oceania), which does not allow us to capture the effects of the concentration outside the domains. Considering your comment, we therefore examined the contribution of G20 consumption to the PM concentration in all the grid squares located at the edges of each domain. Figures SI 11 -16 report the percentage contribution of each G20 country's consumption-based emissions to PM_{2.5} concentrations in the edge grid squares of each of the 6 domains. As can be seen, the largest contribution of Chinese consumption to the PM_{2.5} concentration in a peripheral grid square was about 26%, but here the concentration in the square concerned was only about 5 µg/m³. For all the peripheral grid squares of the Asian domain, the average concentration deriving from Chinese consumption was 0.14 µg/m³. Overall, we confirmed that the impact outside the domain is non-zero, but sufficiently small compared with the impact inside the domain. The following sentences have been added to the main text.

Lines 293 - 306

Although the regional chemical transport model CMAQ³⁹ used for calculating G20 consumption-induced PM_{2.5} concentrations has a spatial resolution of 45-km grid squares for more precise country-by-country analysis, at the expense of detail the world was divided into six domains to reduce the computational burden. This division makes it impossible to analyse the impact of consumption-induced concentrations outside each domain, although, for instance, Zhang et al.⁸ estimated that 2.2% and 2.0% of premature deaths within the US and Canada, respectively, in 2007 were attributable to long-range transboundary pollution from China. We therefore quantified the contribution of G20 consumption-induced concentrations to the concentrations in the grid squares located on the eastern, western, northern and southern edges of each domain (Figures S11-16 in the SI). For example, at the four edges of the Asian region, the largest contribution of China's consumption to the PM_{2.5} concentration was 26%, in a grid square with about 5 µg/m³. Across all peripheral grid squares, the average concentration contributed by China's consumption was 0.14 µg/m³, suggesting that while the impact outside the region is non-zero, it is sufficiently small compared with that inside the region.

Comment 5

I found the discussion on cause and gender distracting and less interesting / tangential. The main story is about international inequities for total health impacts. The study would be stronger to remove this section and instead add material in response to other lackings mentioned in this review.

Response

All the figures and sentences related to this issue have been deleted.

Comment 6

The lifetime projections are a really neat finding, and will make very noteworthy news splashes / press releases. However, they seem flawed and subject to misinterpretation. Allowing only population to grow but not accounting for changes in mortality rates presents an inconsistency. Growing the footprints as a linear scaling with population also seems unfounded. Over a 70 yr lifetime, emissions regulations are going to change the emissions so dramatically that such an assumption seems absurd. Also the spatial distribution would change, and atmospheric transport will change (owing to GHG-driven climate change). In total, the authors have addressed only a small fraction of the components needed to make accurate forecasts, so I recommend not making such

forecasts at all, as the nuances of what is included and what is neglected will be overlooked in summaries. Instead, just use the present day conditions extended over the lifetime, and present results with the phrase: If current pollution, population, and consumption levels remained constant, the lifetime consumption in country X would claim Y lives. This number is much easier to understand.

Response

Since population growth and aging are future social changes involving relatively little uncertainty, we have included them as a factor modifying the current footprint, but agree that they are insufficient in terms of future projections of the footprint. Following your suggestion, we have simply estimated a lifetime consumption footprint based on the current footprint and the average life expectancy in each G20 nation (see Figure 6).

Figure 6: Relationship between life expectancy, PM2.5-related premature deaths induced by lifetime consumption per capita and percentage of foreign deaths in PM2.5 premature death footprint in each G20 nation. Country codes as follows: Argentina (AR), Australia (AU), Brazil (BR), Canada (CA), China (CN), France (FR), Germany (DE), India (IN), Indonesia (ID), Italy (IT), Japan (JP), Mexico (MX), Russia (RU), Saudi Arabia (SA), South Africa (ZA), South Korea (KR), Turkey (TR), United Kingdom (UK) and United States (US).

Comment 7

Throughout —there is little to no discussion of uncertainty. This needs to be addressed and quantified along the following 3 parts of the analysis: emissions estimates, atmospheric modeling, health impact responses. Some of these are going to have uncertainties of at least x2 — thus it seems misleading to present other results to 4 significant figures. Comparison of model estimated PM2.5 concentrations to measurements (in situ or even satellite derived) is required. Comparison of total burden of disease in 2010 with the base case simulation to prior studies is required.

Response

As per your suggestion, we have added an Uncertainty section to the main text (lines 255 - 306). By citing the related references, the uncertainty of the emission inventory map used is now reported, the model-simulated concentrations and the grid concentrations reported in GBD (Global Burden of Disease) 2016 are compared in Figures SI-4 and the reliability of the GBD concentrations is described. However, it was impossible to determine the comprehensive error in the footprint after integrating all the error factors. We would ask your understanding to leave this as a future issue. In addition, the footprint relationship diagram for infant deaths, which reported deaths to 4 digits, has now been rounded to the nearest 10 with a confidence interval of 95% (see Figure 5).

Lines 255 - 306

2.5 Uncertainty

The 95% confidence intervals for the number of premature deaths reported above reflect the uncertainty of the IER (Integrated Exposure-Response) model^{28,32} that determines the relative risk associated with ambient PM_{2.5} exposure. The IER function has been used in many previous footprint studies^{8,13,17,19}. Recently, the developers of the IER function themselves proposed the GEMM (Global Exposure Mortality Model)³³, which improves the relative risk estimation based on cohort studies of ambient PM_{2.5} mortality, covering higher outdoor PM_{2.5} concentrations, which has been a challenge in the IER model. Here, we estimated the premature deaths due to G20 footprints and those of production emissions using the GEMM 5-COD model and in each case compared them with the IER results in terms of the five diseases studied and exposures above 25 years' age targeted by the GEMM. Although there are differences among nations, the GEMM-based footprints are all 1.5 to 2 times larger than the IER footprints (Figure 7). The extent of this increase is consistent with Burnett et al.³³, who confirmed a change in global premature deaths in 2015 from 4.002 million premature deaths based on the IER to 6.889 million deaths with the GEMM 5-COD. As for premature deaths within China, Liu et al.³⁴ report an increase from 0.986 million deaths with the IER to 1.681 million deaths with the GEMM 5-COD in 2010. In all nations except Australia, premature deaths due to production emissions showed a higher increase than for the footprint, confirming the trend that uncertainty in footprint deaths is less affected by the choice of relative risk model.

The other uncertainties derive from the accuracy of the air pollutant emissions inventory. Uncertainty has not been estimated for the emission inventory of EDGAR v4.3.1³⁵ used in this study. However, for the updated EDGAR v4.3.2, uncertainty estimates are provided for air pollutant emissions, based on the estimated uncertainty of the activity data and emission factors for each emission sector, pollutant and country³⁶. The two EDGAR databases are consistent for almost all sources and there is no significant discrepancy in global emissions between them except for NMVOC³⁶. The uncertainties in the EDGAR v4.3.2 emission inventory can therefore be taken as holding for EDGAR v4.3.1, too. The uncertainties for primary PM_{2.5} vary by region, with estimates ranging from 49.4% to 96.8% in 2010, as do those for precursors of secondary particles: of these, NH₃ (185.0% to 294.6%) has the highest uncertainty, while SO_x (12.5% to 48.3%) and NO_x (17.9% to 117.9%) are reported to have relatively small estimation errors³⁶. The present study employs annual average PM_{2.5} concentrations compiled in GBD2016 (Global Burden of Disease)^{37,38} to estimate the total premature deaths in each grid square. In GBD, grid squares lacking ground-based measured PM_{2.5} concentrations are supplemented by the data integration model with information from satellite-based estimates and chemical transport models within a Bayesian hierarchical modelling framework³⁸, which contributes to the uncertainty in our estimate of number of premature deaths in these grid squares.

Although the regional chemical transport model CMAQ³⁹ used for calculating G20 consumption-induced PM_{2.5} concentrations has a spatial resolution of 45-km grid squares for more precise country-by-country analysis, at the expense of detail the world was divided into six domains to reduce the computational burden. This division makes it impossible to analyse the impact of consumption-induced concentrations outside each domain, although, for instance, Zhang et al.⁸ estimated that 2.2% and 2.0% of premature

deaths within the US and Canada, respectively, in 2007 were attributable to long-range transboundary pollution from China. We therefore quantified the contribution of G20 consumption-induced concentrations to the concentrations in the grid squares located on the eastern, western, northern and southern edges of each domain (Figures S11-16 in the SI). For example, at the four edges of the Asian region, the largest contribution of China's consumption to the PM_{2.5} concentration was 26%, in a grid square with about 5 µg/m³. Across all peripheral grid squares, the average concentration contributed by China's consumption was 0.14 µg/m³, suggesting that while the impact outside the region is non-zero, it is sufficiently small compared with that inside the region.

Figure 5: Relationships between consumer countries and impacted countries (“to”) for infant deaths (zero to less than five years old) in PM_{2.5} premature death footprint of G20 nations.

Comment 8

The sudden introduction of GHG implications is unsupported and not well considered. GHG emissions do not well correlate with PM_{2.5} health impacts, because it is possible to have high-carbon low-pollution economies (like the US). See for example Anenberg et al, Sci Reports, 2020. Please just remove this discussion altogether.

Response

Thank you. All related text has been deleted.

Comment 9

308-311: This is an odd conclusion —how is one going to have policy that counters population growth and aging? Regardless, just remove the component of the analysis owing to population projections, as recommended above.

Response

As we have revised the calculation of premature deaths associated with lifetime consumption, all related elements of the discussion have now been deleted.

Responses to Comments by Reviewer 2

Comment 1

First, and foremost, there is no validation on the model results. To be publishable, they need to show that the model adequately captures the concentration levels for PM_{2.5}, especially for the fractions of primary and secondary PM_{2.5} they emphasized in the manuscript. In addition, there are published results about the premature mortality associated with exposure to outdoor PM_{2.5} for countries they focused on here (for examples, Lelieveld et al., *Nature* 2015, 525 (7569), 367–371; Liu et al., *Sci. Total Environ.* 2016, 568, 1253–1262; Hu et al., *Environ. Sci. Technol.* 2017, 51, 950–9959). They should also compare their estimates in the base case to those published results to build confidence.

Response

Thank you for introducing these key papers for comparison. We have now compared our study with previous studies in terms of premature deaths worldwide, and for China and India. Specifically, the values reported in the papers cited below are compared. The simulated concentrations and grid concentrations reported in GBD (Global Burden of Disease) 2016 are also compared. With respect to secondary particles, the simulated-concentrations of several major components are compared with the observation data reported in IMPROVE (<http://vista.cira.colostate.edu/improve/Data/data.htm>), EMEP (<https://projects.nilu.no/ccc/index.html>) and EANET (<https://monitoring.eanet.asia/document/public/index>). See Figure SI 4.

Comment 2

Second, it is unclear how they attributed the premature mortality to primary vs. secondary. Did they apply the fractions of primary and secondary in PM_{2.5} concentrations to the total premature mortality? Or they estimated the premature mortality using the primary and secondary concentrations? In any case, they assumed the same exposure-response functions for the primary and secondary PM_{2.5}. Although it is a ‘standard’ treatment, the novelty of the manuscript emphasizing the ‘secondary’ fraction in the results is not very convincing.

Response

The difference in mortality risk between primary and secondary PM_{2.5} particles has not been taken into account. While some data suggest there may be a discrepancy in risk due to different particle chemical composition, at present there is insufficient evidence to create source-specific exposure-response functions (Shaffer, R.M. et al., *Environmental Health Perspectives*, **127**(10), 105001, doi.org/10.1289/EHP5496 (2019)). The contribution to the number of premature deaths was therefore simply determined using the ratio between primary and secondary particle concentrations. The reason for separating out the impact of secondary particles is to clearly emphasize to the G20 the importance of abatement measures not only for primary particles but also for precursors of secondary particles. In addition, as mentioned in the Introduction section, many previous PM_{2.5} footprint studies focus only on primary particles, and we believe it would be helpful to indicate how many additional premature deaths occur as a result of secondary particles.

Comment 3

Third, it is a little bit outdated to use 2010 as the base case year. In addition, the fourth stage of the method is to calculate the per capita lifetime cost based on the data from January 1 to December 31, 2010, to estimate the total premature deaths in 2050 and 2100. One-year data is used to predict the long-term future impact based on population size changes alone. It is not convincing and has too many uncertain factors to make research meaningful.

Response

Since population growth and aging are future social changes with relatively little uncertainty, we have included them as a factor modifying the current footprint, but agree that they are insufficient in terms of future projections of the footprint. Accepting the concrete suggestion by Reviewer 1, we now simply estimate the lifetime consumption footprint based on the current footprint and average life expectancy in each G20 nation (see Figure 6).

Figure 6: Relationship between life expectancy, PM2.5-related premature deaths induced by lifetime consumption per capita and percentage of foreign deaths in PM2.5 premature death footprint in each G20 nation. Country codes as follows: Argentina (AR), Australia (AU), Brazil (BR), Canada (CA), China (CN), France (FR), Germany (DE), India (IN), Indonesia (ID), Italy (IT), Japan (JP), Mexico (MX), Russia (RU), Saudi Arabia (SA), South Africa (ZA), South Korea (KR), Turkey (TR), United Kingdom (UK) and United States (US).

Minor comments:

Comment 4

Where does the global supply chain data come from, and what data is needed for input in the global multi-regional input-output (MRIO) model?

Response

We used the Eora MRIO model, as now reported in the Method section.

Lines 388 - 400

In Stage 1, the global multiregional input-output (MRIO) model Eora⁵¹ for the year 2010, which incorporates global supply chains among 187 countries, and the emissions inventory map EDGAR v.4.3.1³⁵ were combined using the geographical data integration technique^{52,53}, which enabled the mapping of emissions by sectors and by countries determined by an environmental input-output analysis¹³.

Comment 5

The method part lacks model configuration, model verification, and this part needs to be supplemented, such as the MRIO model, CMAQ model and weather research and forecast model

Response

The version numbers of models and, where relevant dates, have been added to the Method section: Eora MRIO for the year 2010, EDGAR ver. 4.3.1, GFED ver. 4.1s, MEGAN v.2.1, CMAQ ver. 5.2.1, WRF

ver. 3.8.1, MOZART-4. For model verification, we compared the simulated PM_{2.5} concentrations and grid concentrations reported by GBD (Global Burden of Disease) 2016 in Figure SI-4.

Lines 406 - 412

Therefore, aiming to more accurately capture the impact associated with the emitting countries in finer resolution, this study used the regional chemical transport model CMAQ (version 5.2.1)³⁹ as the air quality model and the regional meteorological model WRF (version 3.8.1)⁵⁴, and set the grid size to 0.5° × 0.5° (about 45 km × 45 km) squares. The vertical airspace was divided into 25 layers extending to 100 hPa. The global chemical transport model MOZART-4⁵⁵ was used to calculate concentrations at the boundaries of the calculated region.

Lines 416 - 425

The models computed hourly PM_{2.5} concentrations (over 12 months from January 1 to December 31, 2010) in each grid square using the original emissions map of EDGAR (v.4.3.1) and other emissions map data (GFED (Global Fire Emissions Database) v4.1s⁵⁶ and MEGAN (Model of Emissions of Gases and Aerosols from Nature) v2.1⁵⁷) (below: 'base-case emissions').

Comment 6

How the territorial emission maps are created is not clear.

Response

We created the emission maps using EDGAR ver. 4.3.1 by extracting emissions within the territory of each G20 nation. This is reported in the Methods section.

Lines 398 - 400

The production-based emissions map, showing the direct emissions generated in each G20 nation, was also compiled from the EDGAR emissions map by extracting the emissions within the territory of each nation.

Comment 7

The number for India in Fig. 6 is 44126?

Response

Yes, but it has been rounded to 44100 deaths in what is now Figure 5.

Figure 5: Relationships between consumer countries and impacted countries (“to”) for infant deaths (zero to less than five years old) in PM2.5 premature death footprint of G20 nations.

Comment 8

Are all the numbers reported in the manuscript the median estimates? To be more meaningful, 95% Cis should be included and some discussions should be provided about the uncertainties in the estimates.

Response

We calculated premature deaths based on the relative risk model, IER, formulated with the 50th percentile values of sample data. Following your suggestion, we have now also calculated mortality using the IER based on the 97.5th and 2.5th percentile values, allowing us to specify the 95% confidence interval (CI) of each estimate. Error bars showing the CI have also been added to the figures. We have also verified the uncertainty of the footprint when applying the new relative risk model, GEMM, in response to a suggestion by Reviewer 1 (see Figure 7).

REVIEWER COMMENTS

Reviewer #1 (Remarks to the Author):

The revised manuscript by Nansai et al. is a significant step closer to being publishable in Nat Com., although still not yet. In response to the first round of reviewer comments, they have appropriately removed parts of their original analysis that were overly uncertain or tangential. They have also added requested model evaluation, discussion of uncertainty, and comparison to previous studies. However, they have not yet managed to incorporate the latter into their analysis or conclusions.

For example, their abstract still does not contain a single estimate of uncertainty. The statement that “G20 lifetime consumption of 28 people claims one life” is sure to make headlines. But it needs to be qualified. Is this 14 - 42 people? Or precisely 28 people? In other parts of the manuscript, significant sources of uncertainty are listed, such as more than 100% uncertainty in emissions of several species, and more than 100% uncertainty in health impact calculations. However, the authors fail to integrate these sources of uncertainty into their conclusions. They claim it is “impossible to determine the comprehensive error in the footprint after integrating all the error factors,” which is offensively wrong. At a minimum they could combine their errors in quadrature as a reasonable first order approximation. More rigorously, here is the wikipedia article on error propagation, which is an elementary calculation that I often require in reports from undergraduate students:

https://en.wikipedia.org/wiki/Propagation_of_uncertainty. The partial derivatives of the IER function are easy to calculate. The partial derivatives of their CMAQ simulations are approximated by the model response experiments they have already performed. These could be pieced together to rigorously propagate uncertainties.

I think my concerns about the conclusions of this work being stated too firmly and confidently are well founded. Please just take a look at SI Fig 1. Here is an example of how the authors did respond to review requests for comparison of their model to observations, by including this figure, but also an example of how they made only a minimal attempt to incorporate this comparison into their analysis. The results of this model evaluation are not even mentioned in the main manuscript (they only mention that the evaluation was conducted, but don't state what it showed). So, in case my recommendation isn't clear, here's an example of how they should be thinking about and using this evaluation. The model seems to significantly overestimate PM2.5 in Indonesia, by x2 to x3. Why is this, and does it mean that their estimates of premature deaths here are too high? Similarly, for Mexico they are too low by about a factor of two — does this mean they are underestimating the role of emissions in Mexico associated with US consumption? I realize they scale their total PM2.5 values to the GBD values for the base case, but still the slope of these lines is significant, in that it shows the model response, which they use directly in their calculations. Hence even regions with very tight correlations but biased slopes (e.g.

France, Canada) are of concern. I'm curious about the region of China where they appear to miss high PM2.5 concentrations. Also, it seems they may have issues with natural aerosol in their model, as their model estimates in Saudi Arabia are unphysical low for this part of the world, where annual PM2.5 concentrations should well exceed 100 ug/m³ owing to dust. It seems they also lack fidelity in Argentina, Australia, and Turkey — should these then be omitted from their final results?

I can appreciate that setting up CMAQ simulation on multiple domains around the world is a significant undertaking, and getting the simulations to be reasonable in each location takes a lot of effort. This is precisely why previous studies have used global models for this sort of analysis. But it appears the authors are not quite yet done with their model evaluation, or done with using this type of analysis to inform their conclusions.

Reviewer #2 (Remarks to the Author):

The authors have made great efforts to clarify my concerns. The comments raised by other reviewers have also been appropriately addressed, and the manuscript has been greatly improved. I suggest it to be accepted for publication.

Responses to Comments by Reviewer 1

Comment 1

The revised manuscript by Nansai et al. is a significant step closer to being publishable in Nat Com., although still not yet. In response to the first round of reviewer comments, they have appropriately removed parts of their original analysis that were overly uncertain or tangential. They have also added requested model evaluation, discussion of uncertainty, and comparison to previous studies. However, they have not yet managed to incorporate the latter into their analysis or conclusions. For example, their abstract still does not contain a single estimate of uncertainty. The statement that “G20 lifetime consumption of 28 people claims one life” is sure to make headlines. But it needs to be qualified. Is this 14 - 42 people? Or precisely 28 people?

Response

Thank you for your kind understanding of our previous revisions and the additional analyses. In this revision, our main improvements concern evaluation of CMAQ-modelled concentrations against GBD2016 uncertainty data and comparison of CMAQ-simulated AOD (Aerosol Optical Depth) with observed AOD; we have added our interpretation of the evaluation to the main text by referring to several existing papers on real-world PM_{2.5} data. We have also modified all the values cited in the Abstract to indicate their confidence intervals as follows and tried to keep the wording of the text suitably modest.

Lines 13 – 16

Factoring in the specific life expectancy in each G20 nation, this footprint means that on average the G20 lifetime consumption of about 28 [CI: 24, 33] people claims one life. Lifetime consumption in China has the greatest impact, with 20 [CI: 17, 23] people’s consumption costing one life.

Comment 2

In other parts of the manuscript, significant sources of uncertainty are listed, such as more than 100% uncertainty in emissions of several species, and more than 100% uncertainty in health impact calculations. However, the authors fail to integrate these sources of uncertainty into their conclusions. They claim it is “impossible to determine the comprehensive error in the footprint after integrating all the error factors,” which is offensively wrong. At a minimum they could combine their errors in quadrature as a reasonable first order approximation. More rigorously, here is the wikipedia article on error propagation, which is an elementary calculation that I often require in reports from undergraduate students: https://en.wikipedia.org/wiki/Propagation_of_uncertainty. The partial derivatives of the IER function are easy to calculate. The partial derivatives of their CMAQ simulations are approximated by the model response experiments they have already performed. These could be pieced together to rigorously propagate uncertainties.

Response

We appreciate your suggestion to apply the error propagation equation to estimate the footprint's error as a first-order approximation. In our methodology, the footprint (f) of each country is determined by allocating the total number of premature deaths per grid square (p) by the PM_{2.5} concentration ratio (r) of each country to that square, viz. it can be simply formulated as $f = p \times r$. In line with the error propagation equation, since $df/dr = p$ and $df/dp = r$, the footprint error σ_f can be formulated as:

$$\sigma_f = \{(df/dr \times \sigma_r)^2 + (df/dp \times \sigma_p)^2\}^{(1/2)} = \{(p \times \sigma_r)^2 + (r \times \sigma_p)^2\}^{(1/2)}.$$

Given that p in each grid square is the number of premature deaths calculated from the average GBD2016 concentration and the average IER, and r is the CMAQ-simulated concentration ratio calculated from the average EDGAR emissions, the standard deviation of p and r (σ_p and σ_r) are needed.

The σ_p for each grid square in the world can be estimated using the 95% confidence interval (CI) provided by GBD2016 and the 95% CI for IER. However, our computer environment (workstation) was not capable of producing a sufficiently large data sample to determine a reliable value of σ_r in each grid square in each country by randomly changing the emissions in EDGAR. After much trial and error, we therefore opted to analyse the uncertainty in premature deaths in a grid square (p) and that in concentration ratio (r) in G20 footprints separately. We ask for your understanding of the validity of our approach to tackling this challenge under the given budget constraints, which preclude purchase of additional workstations.

In concreto, we calculated nine different patterns of premature deaths (p) in each grid square, applying each of the three IERs (mean, 95% CI upper and lower bound cases) to the mean, upper and lower bound concentrations in the 95% CI of the GBD2016 data. The results were then aggregated to show the difference in the footprint of each G20 country (the percentage change when the mean concentration is taken as baseline 1). Two other patterns of premature deaths (p) were additionally calculated, one with the 2010 concentration provided in GBD2013, the dust data of which was employed in this study, and the other derived directly from the CMAQ-simulated grid-square concentration. In all, we thus compared five different footprints for each country for a single IER (the left panel of Figure 7 shows the IER (average) case). Here, for all patterns, the concentration ratio (r) for each grid square was estimated from the CMAQ-simulated concentration using the average emissions in EDGAR.

As a result, when applying the IER (average) to the lower-bound case of the GBD2016 concentrations, the footprints become 0.64 (South Africa) to 0.81 (China and India) times lower than those with the average concentration; the average for the G20 countries is 0.72 times. In contrast, at the upper bound concentrations, the footprints varied between 1.11 times (India) and 1.24 times (South Africa), with the average for G20 countries 1.17 times. Using GBD2013 concentrations as well as CMAQ-simulated concentrations leaves all the footprints still within the above range, with the exception of Indonesia. We believe that these multiples adequately represent the uncertainties in the footprints due to the overall uncertainties in the concentration data.

The following sentences were added the revised main text.

Lines 299 – 311

Taking the lower bound of the 95% CI of GBD2016, the footprints become 0.64 times (South Africa) to 0.81 times (China, India) lower, with an average of 0.72 times for the G20 nations (Figure 7, left). With the upper bound concentrations they increase between 1.11 times

(India) and 1.24 times (South Africa), with a G20 average of 1.17 times. On the other hand, the footprint based on the 2010 average concentration in GBD2013¹ with the dust concentration data employed in this study gives a G20 mean of 0.90 times. The gridded concentrations directly calculated by the CMAQ give a G20 mean of 0.95 times, showing similar changes in the footprint as in the case of GBD2013. With both the direct CMAQ and GBD2013, the footprints of the G20 countries are generally within the range of the upper and lower bounds of the CI of GBD2016, which would support the reasonability of allocating total premature deaths in a grid based on GBD2016 concentration data by the consumption-based PM_{2.5} concentration computed by the CMAQ.

On the other hand, we examined the validity of using the CMAQ-simulated concentrations based on the average emissions in EDGAR for the concentration ratio (r) via comparison with the GBD2016 95% CI range and with AOD (Aerosol Optical Depth) observations and by investigating the response of CMAQ-simulated concentrations to the maximum increase and decrease of emissions. We confirmed that CMAQ-simulated concentrations are generally close to the GBD2016 95% CI (Figure S4). Some territories where CMAQ-simulated concentrations proved to be beyond the GBD2016 95% CI showed relatively good agreement with the observed AOD (Figure S13). These assessments point to CMAQ-simulated concentrations not deviating significantly from reality and lying within the range observed in practice.

In territories with a high dust concentration, however, CMAQ was found to have weak reproducibility. To overcome this challenge, we extrapolated the 2010 dust concentrations from GBD2013 to each grid square to improve the reliability of the local value. (GBD2013 specifies grid squares where high dust concentrations (over 50%) are estimated.)

CMAQ-simulated concentrations were also computed by inputting the upper and lower emissions of the 95% CI of EDGAR and these values compared with the concentrations from EDGAR mean emissions. As a result, the CMAQ-simulated concentrations are generally linearly related to the increase or decrease in emissions (Figure S5). We believe that as long as the emission inventory has an overall consistent tendency towards over- or underestimation, this linear response would mean the uncertainty in emissions data has only a minor impact on the concentration ratio (r).

In addition, the CMAQ-simulated concentrations using the average emissions lie roughly in the middle of the CMAQ concentrations calculated from the upper and lower bound emissions. Therefore, as long as the emissions vary randomly and uniformly between the upper and lower cases, the average value of the concentration is expected to approach the concentration with average emissions. Overall, then, we feel the above analysis justifies taking the concentration ratio (r) based on the average emissions used in this study as a representative index for mortality allocation to a given grid square.

Figure 7: (Left) Change in premature deaths footprint when the other PM_{2.5} concentration data (LB16, UB16, ME13, CMAQ) are used for grid-square concentrations to estimate total premature deaths in each square, relative to 1 for premature deaths with the IER (Integrated Exposure-Response) model (50% value) and PM_{2.5} concentration (mean) in GBD2016. LB16 and UB16 are the cases with the lower and upper bound concentration in 95% confidence intervals (CI) of GBD2016, respectively. ME13 is the case with GBD2013 concentration (mean) in 2010 and CMAQ is the case with concentration calculated by the CMAQ model. The cases of IER with upper and lower bounds of the 95% CI are shown in Figure S12.

Figure S4: Comparison of CMAQ-simulated $PM_{2.5}$ concentrations using base-case emission map with GBD2016 concentration data (black dots, Figures (1)-(25)) with the 95% CI range (grey lines); red dots are grid squares with dust concentration indicated in GBD2013; grid squares with a population density over 22.5 persons/ km^2 are shown.

Comparison of CMAQ-simulated $PM_{2.5}$ secondary particle concentrations with observation data in IMPROVE (<http://vista.cira.colostate.edu/improve/Data/data.htm>), EMEP (<https://projects.nilu.no/ccc/index.html>) and EANET (<https://monitoring.eanet.asia/document/public/index>), respectively (Figures (26)-(28))

Figure S13: Comparison of CMAQ-simulated AOD (Aerosol Optical Depth) with observed AOD; red dots are grid squares where dust concentration is indicated in GBD2013; grid squares with population density over 22.5 persons/km² are shown

Figure S5: Comparison of CMAQ-simulated PM_{2.5} concentrations with mean emissions of EDGAR using upper and lower bound emissions in 95% CI of EDGAR (green: with upper bound emissions, blue: with lower bound emissions)

Comment 3

I think my concerns about the conclusions of this work being stated too firmly and confidently are well founded. Please just take a look at SI Fig 1. Here is an example of how the authors did respond to review requests for comparison of their model to observations, by including this figure, but also an example of how they made only a minimal attempt to incorporate this comparison into their analysis. The results of this model evaluation are not even mentioned in the main manuscript (they only mention that the evaluation was conducted, but don't state what it showed). So, in case my recommendation isn't clear, here's an example of how they should be thinking about and using this evaluation. The model seems to significantly overestimate PM_{2.5} in Indonesia, by ×2 to ×3. Why is this, and does it mean that their estimates of premature deaths here are too high? Similarly, for Mexico they are too low by about a factor of two — does this mean they are underestimating the role of emissions in Mexico associated with US consumption? I realize they scale their total PM_{2.5} values to the GBD values for the base case, but still the slope of these lines is significant, in that it shows the model response, which they use directly in their calculations. Hence even regions with very tight correlations but biased slopes (e.g. France, Canada) are of concern. I'm curious about the region of China where they appear to miss high PM_{2.5} concentrations. Also, it seems they may have issues with natural aerosol in their model, as their model estimates in Saudi Arabia are unphysical low for this part of the world, where annual PM_{2.5} concentrations should well exceed 100 µg/m³ owing to dust. It seems they also lack fidelity in Argentina, Australia, and Turkey — should these then be omitted from their final results?

Response

Thank you for pointing out a significant issue. The PM_{2.5} concentrations in GBD2016 are not only based on observations, but also include many estimates obtained statistically using satellite data and chemical transport models. This means the GBD2016 95% CI is fairly broad, which moves us to judge that it would be misleading to compare the CMAQ-simulated concentrations solely with the mean values of GBD2016. As mentioned above, in this revision we include a comparison with the GBD2016 95% CI, as well as a comparison between the CMAQ-simulated AOD by and the observed AOD.

In general, including France, China, Argentina and Australia, the CMAQ-simulated concentrations exhibit values close to the GBD2016 confidence intervals. While in Canada the CMAQ and the GBD2016 95% CI have no overlap in certain concentration ranges, overall the concentrations in this country show relatively good correlation between CMAQ-calculated and observed AOD. We therefore believe the CMAQ-simulated concentrations are not significantly unreliable.

In Mexico and Turkey, however, there are some inconsistencies in terms of high AOD concentrations. This may be due to elevated dust concentrations, as reported in several papers²⁻⁴, although in these countries GBD2013 has no grid squares with high dust ratios. Underestimating the dust ratio may lead to overestimating the footprint of "other factors", but not in the case of the G20 countries, because the footprint of "other factors" comprises all deaths minus the contribution from G20 own consumption, dust, forest fires and biogenic VOCs (volatile organic compounds). These "other factors" are not explicitly dealt with in this paper, however.

The following sentences were added the revised main text.

Lines 539 – 557

The annual PM_{2.5} concentrations simulated by CMAQ generally fall into the 95% CI of the GBD2016 data. Most of the high GBD2016 PM_{2.5} concentrations deviating furthest from the simulated values are affected by dust, which CMAQ has difficulty simulating accurately. Influences of dust were corrected in the following Stage 3. While no grid squares are affected by dust in Mexico or Turkey (Figure S4 in the SI) according to the GBD2013 criteria (PM_{2.5} ≥ 36 µg/m³ and dust fraction ≥ 50%), in both countries PM_{2.5} may still in fact be affected by dust to a certain extent, as has indeed been reported²⁻⁵. In developing the GBD2016 data, ground monitoring data, remote sensing satellite data and chemical transport models were integrated and good

correlations with the GBD2016 data therefore do not necessarily mean better model performance for ambient PM_{2.5} concentrations. Figure S4 in the SI reveals narrower variations in the PM_{2.5} concentrations from the GBD2016 data than the values simulated by CMAQ. However, such features are not found in comparisons of the observed⁶ and CMAQ-simulated AOD (Aerosol Optical Depth) (Figure S13 in the SI), suggesting that variations in actual ambient PM_{2.5} concentrations are greater than in the GBD2016 data and were reproduced well by CMAQ. In particular, large deviations in Indonesia suggest possible underestimations in the GBD2016 data. Correlations in SO₄²⁻, NO₃⁻, and NH₄⁺ concentrations indicate good performance of CMAQ on secondary components of PM_{2.5}.

For Indonesia, the CMAQ-simulated concentrations are higher than the GBD2016 95% CI, but the observed and CMAQ-based AOD data are similar, implying the existence of high concentrations in Indonesia. In addition, several papers⁷⁻⁹ with real-world observations in urban areas of Indonesia report annual averages such as 40 – 50 µg/m³ in 2001-2007, 33 µg/m³ in 2014 and 27 – 69 µg/m³ in 2007.

The following sentences were added the revised main text.

Lines 311 – 317

In the case of Indonesia, however, the footprint with the CMAQ (1.39 times) exceeds the upper bound case. Even at the upper bound of the CI there is only one grid square with a PM_{2.5} concentration over 30 µg/m³ among Indonesian grid squares with a population density of 22.5 persons/km² or more (Figure S4 in the SI), despite some studies observing concentrations as high as 40 – 50 µg/m³ in 2001-2007⁷, 33 µg/m³ in 2014⁹ and 27 – 69 µg/m³ in 2007⁸ in Bandung City. This suggests possible underestimation of Indonesia's footprint.

In Figure S4, by coloring grid squares with high dust concentrations (>50%) in GBD2013 red, we have made clear that the high concentrations in Saudi Arabia and China are due to dust. As you point out, CMAQ does not seem to adequately simulate these high dust levels. In this revision we have therefore extrapolated the dust concentrations reported by GBD2013 into each grid square to improve the reliability of these concentrations before the footprints were calculated.

Comment 4

I can appreciate that setting up CMAQ simulation on multiple domains around the world is a significant undertaking, and getting the simulations to be reasonable in each location takes a lot of effort. This is precisely why previous studies have used global models for this sort of analysis. But it appears the authors are not quite yet done with their model evaluation, or done with using this type of analysis to inform their conclusions.

Response

The model evaluation and its interpretation, discussed above, have been added to the Results and the Methods sections. Also, in the Discussion section, we now refer to the technical measures required to reduce the uncertainty of the footprint and cite the importance of G20 countries addressing PM_{2.5} health impacts without delay, even though there still remains a degree of uncertainty in the footprints.

The following sentences were added the revised main text.

Lines 455 – 471

At the upper and lower bounds of uncertainty in the PM_{2.5} concentration data, the G20 country footprints vary by an average of 0.72 and 1.17 times, respectively, which seems less severe than the impact of the choice of relative risk model described above. To significantly reduce the uncertainty of the PM_{2.5} concentration data would require global expansion of ground monitoring points for PM_{2.5} concentrations using universally standard measurement methods, but this is prohibited by both time and cost constraints. There is therefore little likelihood of any major decrease in the uncertainty of the footprints in the near term. The critical actions by policymakers called for here are that the G20 nations agree on a unique choice of footprint within the given range of uncertainty and that the G20 takes joint action as soon as possible to reduce the number of premature deaths due to its consumption. If these nations agreed on their smallest footprint within the uncertainty, it would be 0.78 times the mean case, which excludes 439 thousand deaths from the mean G20 footprint. Yet, long-term postponement of reduction action means G20 consumption will continue to result in more premature deaths than the number of deaths excluded. We believe the G20's responsibility for its footprint includes the responsibility to choose and agree on a single figure without delay, within the limits of its uncertainty.

References

- 1 Murray, C. J. L. Global, regional, and national age–sex specific all-cause and cause-specific mortality for 240 causes of death, 1990–2013: a systematic analysis for the Global Burden of Disease Study 2013. *The Lancet* **385**, 117-171, doi:10.1016/s0140-6736(14)61682-2 (2015).
- 2 Ginoux, P., Prospero, J. M., Gill, T. E., Hsu, N. C. & Zhao, M. Global-scale attribution of anthropogenic and natural dust sources and their emission rates based on MODIS Deep Blue aerosol products. *Reviews of Geophysics* **50**, doi:10.1029/2012rg000388 (2012).
- 3 Kutralam-Muniasamy, G., Perez-Guevara, F., Martinez, I. E. & Chari, S. V. Particulate matter concentrations and their association with COVID-19-related mortality in Mexico during June 2020 Saharan dust event. *Environ Sci Pollut Res Int*, doi:10.1007/s11356-021-14168-y (2021).
- 4 Wakamatsu, S. *et al.* A Comparative Study of Urban Air Quality in Megacities in Mexico and Japan: Based on Japan-Mexico Joint Research Project on Formation Mechanism of Ozone, VOCs and PM_{2.5}, and Proposal of Countermeasure Scenario. (JICA Research Institute, Tokyo, 2017).
- 5 Gómez-Losada, Á. & Pires, J. C. M. Estimation of Particulate Matter Contributions from Desert Outbreaks in Mediterranean Countries (2015–2018) Using the Time Series Clustering Method. *Atmosphere-Basel* **12**, doi:10.3390/atmos12010005 (2020).
- 6 Levy, R., Hsu, C., et al. *MODIS Atmosphere L2 Aerosol Product. NASA MODIS Adaptive Processing System*, <http://dx.doi.org/10.5067/MODIS/MOD04_L2.061> (2015).
- 7 Lestari, P. & Mauliadi, Y. D. Source apportionment of particulate matter at urban mixed site in Indonesia using PMF. *Atmos Environ* **43**, 1760-1770, doi:10.1016/j.atmosenv.2008.12.044 (2009).
- 8 Permadi, D. A., Kim Oanh, N. T. & Vautard, R. Integrated emission inventory and modeling to assess distribution of particulate matter mass and black carbon composition in Southeast Asia. *Atmos Chem Phys* **18**, 2725-2747, doi:10.5194/acp-18-2725-2018 (2018).
- 9 WHO. *Ambient (outdoor) air pollution database, by country and city*, <<https://www.who.int/data/gho/data/themes/air-pollution/who-air-quality-database/2016>> (2016).

REVIEWERS' COMMENTS

Reviewer #1 (Remarks to the Author):

The authors have done a commendable job of taking the reviewer suggestions into consideration. They have added additional analysis to estimate uncertainty, using reasonable assumptions for computational expediency, and furthermore they have incorporated this uncertainty into their subsequent discussion, analysis, and presentation of results. They make a good case, though the consideration of multiple PM2.5 estimates as well as comparison to AOD, that their model estimates are reasonable in many countries, and they fixed an issue with dust. I still wonder about Argentina — comparison to AOD wasn't super convincing there (unlike Australia, where it was very convincing). There could be significant dust influence in Argentina as well. I don't think this warrants any further review, but would urge the authors to consider a comment or two in the manuscript about this region.

Response to Comment by Reviewer 1

Comment 1

The authors have done a commendable job of taking the reviewer suggestions into consideration. They have added additional analysis to estimate uncertainty, using reasonable assumptions for computational expediency, and furthermore they have incorporated this uncertainty into their subsequent discussion, analysis, and presentation of results. They make a good case, though the consideration of multiple PM2.5 estimates as well as comparison to AOD, that their model estimates are reasonable in many countries, and they fixed an issue with dust. I still wonder about Argentina — comparison to AOD wasn't super convincing there (unlike Australia, where it was very convincing). There could be significant dust influence in Argentina as well. I don't think this warrants any further review, but would urge the authors to consider a comment or two in the manuscript about this region.

Response

Thank you for your appreciation of our previous revisions and additional analyses. The difference between the simulated and observed AOD in Argentina is that the concentration range showing an underestimate of the simulated AOD may be due to desert dust, as you point out. On the other hand, the concentration range with the positive bias in the simulated AOD can be interpreted as an overstatement of the impact of the Amazon forest fires in 2010, the minimum year for the calculation, in the emission inventory.

The following two papers (Dust and forest fires in Argentina) are newly cited and the following text has been added to the main text.

65) Gassó, S. et al. A combined observational and modeling approach to study modern dust transport from the Patagonia desert to East Antarctica. *Atmos Chem Phys* 10, 8287-8303, doi:10.5194/acp-10-8287-2010 (2010).

71) Thornhill, G. D., Ryder, C. L., Highwood, E. J., Shaffrey, L. C. & Johnson, B. T. The effect of South American biomass burning aerosol emissions on the regional climate. *Atmos Chem Phys* 18, 5321-5342, doi:10.5194/acp-18-5321-2018 (2018).

Lines 542 – 545
Exceptional positive biases of AOD in Argentina may be due largely to biomass-burning emissions and transport from the Amazon, where there were intense forest fires in 2010, to northern Argentina ⁷¹